# Oblique Bayesian Additive Regression Trees

**Paul-Hieu V. Nguyen**                                             *pvnguyen5@wisc.edu*
*Department of Statistics*
*University of Wisconsin–Madison*

**Ryan Yee**                                                       *ryee2@wisc.edu*
*Department of Statistics*
*University of Wisconsin–Madison*

**Sameer K. Deshpande**                                   *sameer.deshpande@wisc.edu*
*Department of Statistics*
*University of Wisconsin–Madison*

**Reviewed on OpenReview:** *https://openreview.net/forum?id=l4Qnj4tHBx*

## Abstract

Current implementations of Bayesian Additive Regression Trees (BART) are based on axis-aligned decision rules that recursively partition the feature space using a single feature at a time. Several authors have demonstrated that oblique trees, whose decision rules are based on linear combinations of features, can sometimes yield better predictions than axis-aligned trees and exhibit excellent theoretical properties. We develop an oblique version of BART that leverages a data-adaptive decision rule prior that recursively partitions the feature space along random hyperplanes. Using several synthetic and real-world benchmark datasets, we systematically compared our oblique BART implementation to axis-aligned BART and other tree ensemble methods, finding that oblique BART was competitive with — and sometimes much better than — those methods.

## 1 Introduction

### 1.1 Motivation

Tree-based methods like CART (Breiman et al., 1984), random forests (RF; Breiman, 2001), and gradient boosted trees (GBT; Friedman, 2001) are extremely popular and effective machine learning algorithms. They are simple to train and typically deliver accurate predictions (Hastie & Friedman, 2005, §9.2).

Compared to these models, the Bayesian Additive Regression Trees (BART; Chipman et al., 2010) model is used much less frequently in the wider machine learning community. Like RF and GBT, BART approximates unknown functions using regression tree ensembles. But unlike RF and GBT, BART is based on a fully generative probabilistic model, which facilitates natural uncertainty quantification (via the posterior) and allows it to be embedded within more complex statistical models. For instance, BART has been extended to models for survival (Sparapani et al., 2016; Linero et al., 2022) and semi-continuous (Linero et al., 2020) outcomes; conditional density regression (Orlandi et al., 2021; Li et al., 2023b); non-homogeneous point process data (Lamprinakou et al., 2023); regression with heteroskedastic errors (Pratola et al., 2020); and integrating predictions from multiple models (Yannotty et al., 2024a;b). BART and its extensions typically return accurate predictions and well-calibrated uncertainty intervals without requiring users to specify the functional form of the unknown function and without hyperparameter tuning. Its ease-of-use and generally excellent performance makes BART an attractive "off-the-shelf" modeling tool, especially for estimating heterogeneous causal effects (Hill, 2011; Dorie et al., 2019; Hahn et al., 2020).

Virtually every implementation of RF, GBT, and BART utilizes axis-aligned regression trees, which partition the underlying predictor space into rectangular boxes (Figure 1a) and correspond to piecewise constant step functions. Although step functions (i.e., regression trees) are universal function approximators, several authors have proposed modifications to make regression trees even more expressive and flexible. One broad class of modifications replaces the constant output in each leaf node with simple parametric models (e.g., Quinlan, 1992; Landwehr et al., 2005; Chan & Loh, 2004; Künzel et al., 2022) or nonparametric models (e.g., Gramacy & Lee, 2008; Starling et al., 2020; Maia et al., 2024). Another broad class involves the use of "oblique" decision rules, which allow trees to partition the predictor space along arbitrary hyperplanes (see, e.g., Figure 1b). As we detail in Section 2.1, several authors have found that oblique trees and ensembles thereof often outperform their axis-aligned counterparts in terms of prediction accuracy. For instance, Bertsimas & Dunn (2017) reported oblique decision trees can outperform CART by 7% while Breiman (2001) noted that an oblique version of RF can improve performance by 3%. Beyond these empirical results, Cattaneo et al. (2024) recently demonstrated that oblique decision trees and their ensembles can sometimes obtain the same convergence rates as neural networks. To the best of our knowledge, however, there has been no systematic exploration of oblique trees in the Bayesian setting.

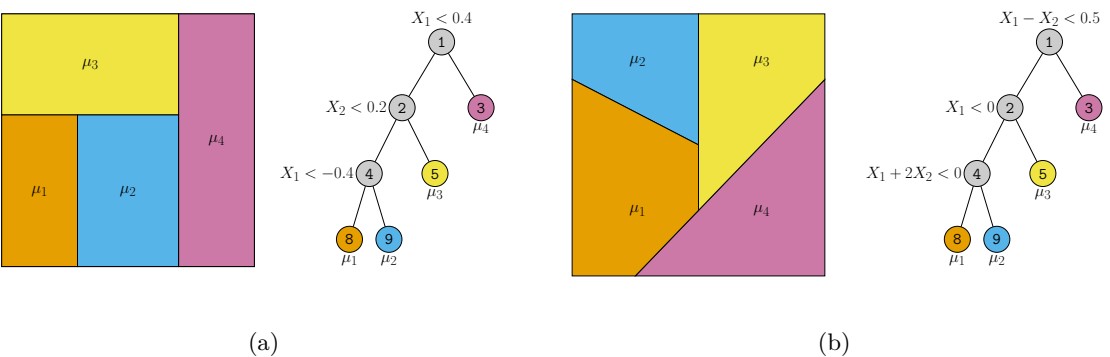

(a)                                                                      (b)

Figure 1: Example of step functions defined over $[-1, 1]^2$ and their corresponding axis-aligned (a) and oblique (b) regression tree representations.

## 1.2 Our contributions

In this work, we introduce obliqueBART and systematically compare its predictive ability to the original BART model and several other tree-based methods. As a preview, consider the functions in Figures 2a and 2d, whose discontinuities are not closely aligned with the coordinate axes. Although axis-aligned BART captures the general shape of the decision boundaries, it requires very deep trees to do so and inappropriately smoothes over the functions' sharp discontinuities (Figures 2b and 2e). ObliqueBART, in sharp contrast, recovers the boundaries much more precisely and with shallower trees (Figures 2c and 2f). We will return to these examples in Section 4.1.

The rest of the paper is organized as follows. In Section 2, we review the literature on oblique trees and briefly review the basic axis-aligned BART model. In Section 3, we introduce our prior on oblique decision rules and discuss our implementation of obliqueBART. We compare the performance of obliqueBART to axis-aligned BART, several popular tree-based machine learning models, and other oblique ensembles across a range of synthetic and benchmark datasets in Section 4. We discuss potential extensions and further refinements in Section 5.

## 2 Background

Before reviewing existing oblique tree models (Section 2.1) and BART (Section 2.2), we introduce some notation and briefly review RF and GBT. For simplicity, we focus on the standard nonparametric regression problem with continuous predictors: given $n$ observations $(\boldsymbol{x}_1, y_1), \ldots, (\boldsymbol{x}_n, y_n)$ with $\boldsymbol{x}_i \in [-1, 1]^p$ from the

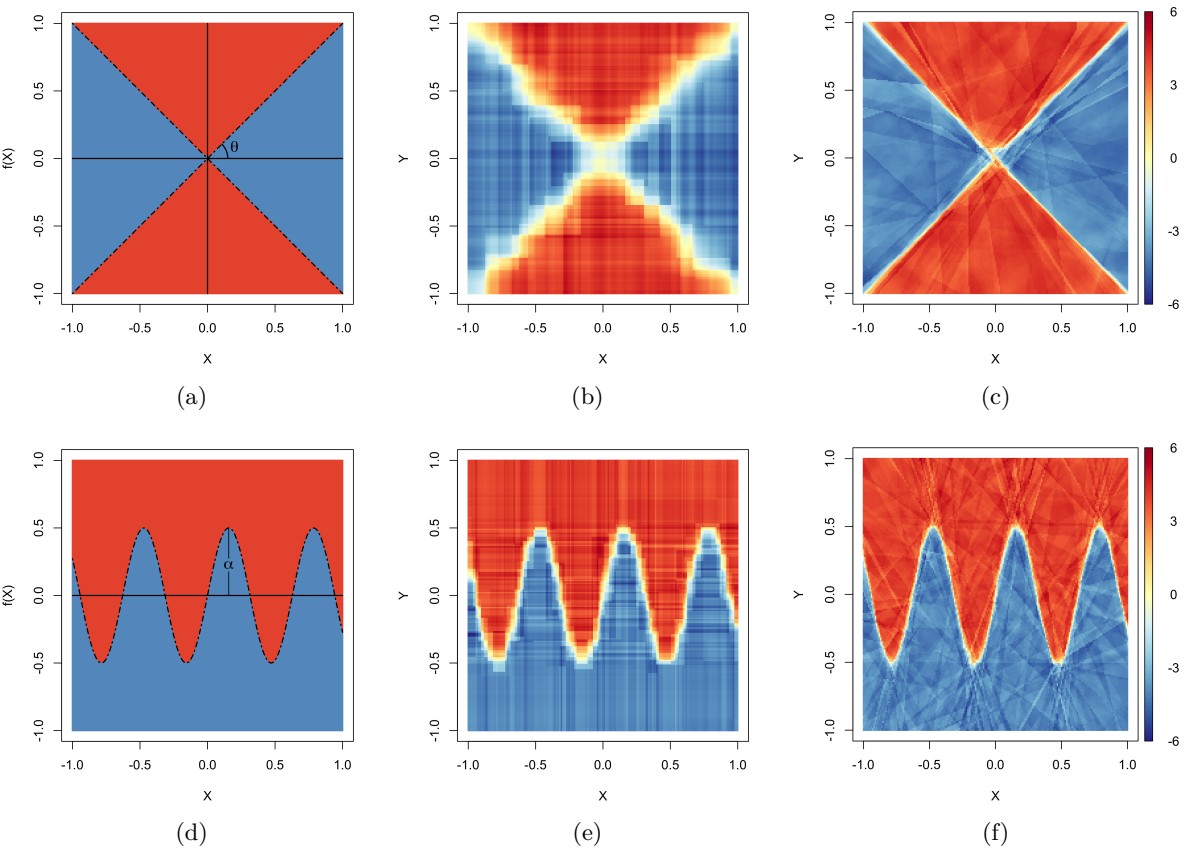

Figure 2: True function (a,d), axis-aligned BART estimate (b,e), and obliqueBART estimate (c,f).

model $y \sim \mathcal{N}\left(f(\boldsymbol{x}), \sigma^2\right)$, we wish to recover the unknown function $f$ and residual variance $\sigma^2$. A *decision tree* over $[-1, 1]^p$ is a pair $(\mathcal{T}, \mathcal{D})$ containing (i) a binary tree $\mathcal{T}$ that contains several terminal or *leaf* nodes and several non-terminal or *decision* nodes and (ii) a collection of decision rules $\mathcal{D}$, one for each decision node in $\mathcal{T}$. Axis-aligned decision rules take the form $\{X_j < c\}$ while oblique decision rules take the form $\{\phi^\top \boldsymbol{x} < c\}$ where $\phi \in \mathbb{R}^p$. Notice that axis-aligned rules form a subset of all oblique rules.

Given a decision tree $(\mathcal{T}, \mathcal{D})$, for each point $\boldsymbol{x} \in [-1, 1]^p$, we can trace a path from the root to a unique leaf as follows. For axis-aligned trees (resp. oblique trees), starting from the root, whenever the path reaches a node with decision rule $\{X_j < c\}$ (resp. $\{\phi^\top \boldsymbol{x} < c\}$), the path proceeds to the left if $x_j < c$ (resp. $\phi^\top \boldsymbol{x} < c$) and to the right otherwise. We let $\ell(\boldsymbol{x}; \mathcal{T}, \mathcal{D})$ denote the leaf reached by $\boldsymbol{x}$'s decision-following path. By associating a scalar $\mu_\ell$ to each leaf in $\mathcal{T}$, the *regression tree* $(\mathcal{T}, \mathcal{D}, \mathcal{M})$ represents a piecewise constant step function over $[-1, 1]^p$. We use $\mathcal{M}$ to denote the set of all $\mu_\ell$'s associated with tree $\mathcal{T}$. Given a regression tree $(\mathcal{T}, \mathcal{D}, \mathcal{M})$, we denote the evaluation function that returns the scalar associated to the leaf reached by $\boldsymbol{x}$ as $g(\boldsymbol{x}; \mathcal{T}, \mathcal{D}, \mathcal{M}) = \mu_{\ell(\boldsymbol{x}; \mathcal{T}, \mathcal{D})}$.

In the nonparametric regression problem, CART estimates a single regression tree such that $f(\boldsymbol{x}) \approx g(\boldsymbol{x}; \mathcal{T}, \mathcal{D}, \mathcal{M})$ using an iterative approach. After initializing $\mathcal{T}$ as the root node, CART visits every current leaf node, finds an optimal decision rule for that leaf, and attaches two children to the leaf, turning it into a decision node. This process continues until each leaf contains a single observation. Typically CART trees are pruned back according to a cross-validation criterion. At each step, CART exhaustively searches for the optimal decision rule using a variance reduction criteria. See Hastie & Friedman (2005, §9.2) for an overview of the CART algorithm.

RF, GBT, and BART instead learn a collection (or *ensemble*) of $M$ regression trees $\mathcal{E} = \{(\mathcal{T}_m, \mathcal{D}_m, \mathcal{M}_m)\}$ such that $f(\boldsymbol{x}) \approx \sum_{m=1}^{M} g(\boldsymbol{x}; \mathcal{T}_m, \mathcal{D}_m, \mathcal{M}_m)$. These methods differ significantly in how they learn $\mathcal{E}$.

RF estimates each tree in $\mathcal{E}$ using independent bootstrap sub-samples of the training data. Each tree in an RF ensemble is trained using an iterative procedure similar to CART. However, instead of exhaustively searching over $p$ features, at each iteration, RF identifies the optimal rule over a random subset of features. When the number of training observations $n$ is very large, each tree-growing iteration of RF can be slow, as it involves searching over all possible cutpoints for the randomly selected features. The Extremely Randomized Trees (ERT) procedure of Geurts et al. (2006) overcomes this limitation by randomizing both the splitting variable and cutpoint. That is, in each iteration, ERT draws a small collection of splitting variable and cutpoint pairs and then finds the optimal rule among this small collection. ERT also differs from RF in that each tree is trained using the whole sample, rather than independent bootstrap re-samples. Typically, the trees in an RF or ERT ensemble are un-pruned.

GBT, on the other hand, trains the trees sequentially, with each tree trained to predict the residuals based on all preceding trees. Specifically, for each $m$, GBT trains $(\mathcal{T}_m, \mathcal{D}_m, \mathcal{M}_m)$ to predict the residuals $y_i - \sum_{m'=1}^{m-1} g(\boldsymbol{x}_i; \mathcal{T}_{m'}, \mathcal{D}_{m'}, \mathcal{M}_{m'})$. Unlike with CART, RF, and ERT, when running GBT, practitioners conventionally fix the tree to have shallow depth. Although BART shares some algorithmic similarities with ERT and GBT, there are crucial differences, which we will exploit when developing obliqueBART.

## 2.1 Inducing oblique trees

Conceptually, extending CART to use oblique rules seems straightforward: one need only find an optimal direction $\phi$ and cutpoint $c$ that most decreases the variance of the observed responses that reach a given node. Unfortunately, the resulting optimization problem is extremely challenging; in the case of classification, the problem is NP-complete (Heath et al., 1993, Theorem 2.1). Consequently, researchers rely on heuristics to build oblique trees. These heuristics fall into three broad classes. The first class of heuristics are model-based: at each decision node, one selects $\phi$ after fitting an intermediate statistical model and then finds an optimal cutpoint for the feature $\phi^\top \boldsymbol{x}$. For instance, Menze et al. (2011) set $\phi$ to be the parameter estimated by fitting a ridge regression at each decision node. Zhang & Suganthan (2014) and Rainforth & Wood (2015) instead performed linear discriminant, principal component, and canonical correlation analyses at each decision node to determine $\phi$. Compared to the axis-aligned random forests, ensembles of these model-based oblique trees respectively improved prediction accuracy by 20%, 2.9%, and 28.7%.

The second class of heuristics involve randomization: instead of searching over all possible directions $\phi$, one draws $D$ random directions $\phi_1, \ldots, \phi_D$; computes $D$ new features $\phi_1^\top \boldsymbol{x}, \ldots \phi_D^\top \boldsymbol{x}$; and then identifies the optimal axis-aligned rule among these new features. Notable examples of randomization-based approaches are Breiman (2001), Blaser & Fryzlewicz (2016), Tomita et al. (2020), and Li et al. (2023a), which differ primarily in the distribution used to draw the candidate directions $\phi_1, \ldots, \phi_D$. Breiman (2001), for instance, restricted the $\phi_d$'s to have $k$ non-zero elements drawn uniformly from $[-1, 1]$. The resulting random combination forests algorithm had decreased out-of-sample errors by 2-3% compared to the axis-aligned random forests. By allowing a variable number of non-zero entries in $\phi_d$ but restricting those entries to $\pm 1$, Tomita et al. (2020) obtained similar performance gains.

Methods based on the first two classes of heuristics estimate or propose new oblique rules in each iteration of the tree growing procedure. An alternative strategy involves pre-computing a large number of feature rotations or projections and then training a standard axis-aligned model using the newly-constructed features. Rodriguez et al. (2006), for instance, combine a PCA pre-processing step with the standard random forest algorithm. Blaser & Fryzlewicz (2016) instead generate a large number of randomly rotated features before building a random forest ensemble. Their random rotation random forest method outperformed axis-aligned methods in about two-thirds of their experiments.

Before proceeding, it is worth mentioning Bertsimas et al. (2021) does not utilize model- or randomization-based heuristics to grow oblique trees. They instead find optimal oblique decision rules using mixed-integer programming. Their procedure, however, restricts the depth and structure of the tree $\mathcal{T}$ *a priori*, is generally not scalable beyond depth three, and led to small increases in the predictive $R^2$.

## 2.2 Review of BART

Like RF, ERT, and GBT, BART expresses the unknown regression function $f(\boldsymbol{x})$ with an ensemble of trees $\mathcal{E} = \{(\mathcal{T}_m, \mathcal{D}_m, \mathcal{M}_m)\}$. Unlike these methods, which learn only a single ensemble, BART computes an entire posterior distribution over tree ensembles. Since the posterior is analytically intractable, BART uses Markov chain Monte Carlo (MCMC) to simulate posterior samples.

### 2.2.1 The BART prior

Key to BART's empirical success is its regularizing prior over the regression tree ensemble. BART models each tree $(\mathcal{T}_m, \mathcal{D}_m, \mathcal{M}_m)$ as *a priori* independent and identically distributed. We can describe the regression tree prior compositionally by explaining how to sample from it. First, we draw the tree structure $\mathcal{T}$ by simulating a branching process. Starting from the root node, which is initially treated as a terminal node at depth 0, whenever a terminal node at depth $d$ is created, the process attaches two child nodes to it with probability $\alpha(1 + d)^{-\beta}$. Then, at each decision node, decision rules of the form $\{X_j < c\}$ are drawn conditionally on the rules at the nodes' ancestors. Drawing a decision rule involves (i) uniformly selecting the splitting axis $j$; (ii) computing the interval of valid values of $X_j$ at the current tree node; and (iii) drawing the cutpoint $c$ uniformly from this interval. The set of valid $X_j$ values at any node is determined by the rules at the node's ancestors in $\mathcal{T}$. For instance in Figure 1a, if we were to draw a decision rule at node 5, $[-1, 0.4]$ is the set of valid $X_1$ values and $[0.2, 1]$ is the set of valid $X_2$ values. Finally, conditionally on $\mathcal{T}$, the leaf outputs in $\mathcal{M}$ are drawn independently from a $\mathcal{N}\left(0, \tau^2/M\right)$ distribution.

Chipman et al. (2010) completed their prior by specifying $\sigma^2 \sim \text{Inv. Gamma}\left(\nu/2, \nu\lambda/2\right)$. They further recommended default values for each prior hyperparameter. They suggested setting $\alpha = 0.95$ and $\beta = 2$, so that the branching process prior concentrates considerable prior probability on trees of depth less than 5. For any $\boldsymbol{x}$, the marginal prior for $f(\boldsymbol{x})$ is $\mathcal{N}\left(0, \tau^2\right)$. Chipman et al. (2010) recommended setting $\tau$ so that this marginal prior places 95% probability over the range of the observed data. They similarly recommended setting $\nu = 3$ and $\lambda$ so that there was 90% prior probability on the event that $\sigma$ was less than the standard deviation of the observed outcome. Finally, they recommended setting $M = 200$. Although some of these choices are somewhat *ad hoc* and data-dependent, they have proven tremendously effective across a range of datasets.

### 2.2.2 Sampling from the BART posterior

To simulate draws from the tree ensemble posterior, Chipman et al. (2010) introduced a Metropolis-within-Gibbs sampler. In each iteration, the sampler sweeps over the entire ensemble and sequentially updates each regression tree $(\mathcal{T}_m, \mathcal{D}_m, \mathcal{M}_m)$ conditionally fixing the remaining $M - 1$ trees. Each regression tree update consists of two steps. First, a new decision tree $(\mathcal{T}, \mathcal{D})$ is drawn from its conditional distribution given the data, $\sigma$, and all other regression trees in the ensemble. Then, new leaf outputs $\mathcal{M}$ are drawn conditionally given the new decision tree, the data, $\sigma$, and the remaining regression trees. In the standard nonparametric regression setting, the leaf outputs are conditionally independent given the tree structure, which allows us to draw $\mathcal{M}$ with standard normal-normal conjugate updates. Most implementations of BART update the decision tree $(\mathcal{T}, \mathcal{D})$ by first drawing a new decision tree from a proposal distribution and accepting that proposal with a Metropolis-Hastings step. The simplest proposal distribution involves randomly growing or pruning $(\mathcal{T}, \mathcal{D})$ with equal probability.

In each MCMC iteration, BART updates each regression tree *conditionally* fixing the other trees in the ensemble $\mathcal{E}$. Consequently, each individual regression tree in a BART ensemble does not attempt to estimate the true regression function well. Instead, like GBT, each tree is trained to fit a *partial residual* based on the fits of other tres. However, unlike GBT, which trains trees sequentially, each tree in the BART ensemble is dependent on every other tree *a posteriori*. This is in sharp contrast to RF and ERT, which independently train each tree to predict the outcome using a bootstrap sample (RF) or full training data (ERT).

BART is, nevertheless, similar to ERT, insofar as both methods grow trees using *random* decision rules. Recall that at each iteration, ERT grows a tree by selecting the best decision rule among a small collection of random proposals. BART, on the other hand, uses a Metropolis-Hastings step to accept or reject a single

random proposal. As a result, BART can sometimes grow a tree with a rule that does not significantly improve overall fit to data or remove a split that does. Thus, unlike CART, RF, ERT, and GBT, we cannot regard the decision rules appearing in a BART ensemble as optimal in any sense.

## 3 BART with oblique decision rules

Suppose that we observe $n$ pairs $(\boldsymbol{x}_1, y_1), \ldots, (\boldsymbol{x}_n, y_n)$ of $p$-dimensional vectors of predictors $\boldsymbol{x}$ and scalar outcomes $y$. For regression problems, we model $y \sim \mathcal{N}\left(f(\boldsymbol{x}), \sigma^2\right)$ and for binary classification problems, we model $\mathbb{P}(y = 1) = \Phi(f(\boldsymbol{x}))$ where $\Phi$ is the cumulative distribution function for a standard normal distribution. Further suppose that we have $p_{\mathrm{cont}}$ continuous predictors and $p_{\mathrm{cat}} = p - p_{\mathrm{cont}}$ categorical predictors. Without loss of generality, we will arrange continuous predictors first; assume that all continuous predictors lie in interval $[-1, 1]^{p_{\mathrm{cont}}}$; and that the $j$-th categorical predictor lies in a discrete set $\mathcal{K}_j$. That is, we assume that all predictor vectors $\boldsymbol{x} = (\boldsymbol{x}_{\mathrm{cont}}^\top, \boldsymbol{x}_{\mathrm{cat}}^\top)^\top$ lie in the product space $[-1, 1]^{p_{\mathrm{cont}}} \times \mathcal{K}_1 \times \cdots \mathcal{K}_{p_{\mathrm{cat}}}$.

ObliqueBART expresses the unknown regression function $f(\boldsymbol{x})$ with an ensemble of $M$ regression trees $\mathcal{E} = \{(\mathcal{T}_1, \mathcal{D}_1, \mathcal{M}_1), \ldots, (\mathcal{T}_M, \mathcal{D}_M, \mathcal{M}_M)\}$ whose decision rules take the form $\{\phi^\top \boldsymbol{x}_{\mathrm{cont}} < c\}$ or $\{X_{p_{\mathrm{cont}}+j} \in \mathcal{C}\}$ where $j \in \{1, \ldots, p_{\mathrm{cat}}\}$ and $\mathcal{C} \subseteq \mathcal{K}_j$. Formally, this involves specifying a prior and computing a posterior over $\mathcal{E}$, from which we can approximately sample using MCMC. Like in the original BART model, we model the individual trees $(\mathcal{T}_m, \mathcal{D}_m, \mathcal{M}_m)$ as *a priori* i.i.d. Similarly, we simulate posterior samples using a Metropolis-within-Gibbs sampler that updates each regression tree one-at-a-time while fixing the others. The main differences between obliqueBART and the original, axis-aligned BART are the decision rule prior and the conditional regression tree updates.

### 3.1 The obliqueBART prior

For obliqueBART, we adopt exactly the same priors for $\mathcal{T}$ and $\mathcal{M}$ as Chipman et al. (2010). That is, we specify the same branching process prior for $\mathcal{T}$ and independent normal priors for outputs in $\mathcal{M}$ with the same default hyperparameters described in Section 2.2. We specify the obliqueBART decision rule prior implicitly, by describing how to draw a rule at a non-terminal node in a tree $\mathcal{T}$. To this end, suppose we are at internal node nx in $\mathcal{T}$ and that we have drawn rules at all of nx's ancestors. With probability $p_{\mathrm{cat}}/p$, we draw a categorical decision rule of the form $\{X_{p_{\mathrm{cat}}+j} \in \mathcal{C}\}$ where $\mathcal{C} \subset \mathcal{K}_j$ and $j \in \{1, \ldots, p_{\mathrm{cat}}\}$ and with probability $p_{\mathrm{cont}}/p$, we draw a continuous decision rule of the form $\{\phi^\top \boldsymbol{x}_{\mathrm{cont}} < c\}$.

Drawing a categorical decision rule involves (i) selecting the splitting variable index $j$; (ii) computing the set $\mathcal{A}$ of valid values of the $j$-th categorical predictor $X_{p_{\mathrm{cont}}+j}$; and (iii) forming a subset $\mathcal{C} \subset \mathcal{A}$ by randomly assigning each element of $\mathcal{A}$ to $\mathcal{C}$ with probability $1/2$. The set $\mathcal{A}$ is determined by the decision rules of nx's ancestors in the tree. Although Chipman et al. (1998) initially used such categorical decision rules in their Bayesian CART procedure, most implementations of BART do not use this prior and instead one-hot encode categorical features. A notable exception is Deshpande (2024), who demonstrated these categorical decision rules often produced more accurate predictions than one-hot encoding.

Drawing a continuous decision rule involves (i) drawing a random vector $\phi$; (ii) computing the interval of valid values of $\phi^\top \boldsymbol{x}_{\mathrm{cont}}$ at nx; and (iii) drawing the cutpoint $c$ uniformly from that interval. Both Breiman (2001) and Tomita et al. (2020) recommend the use of sparse $\phi$'s when inducing oblique decision trees. While Breiman (2001) fixed the number of non-zero elements of $\phi$, Tomita et al. (2020) demonstrated that allowing the number of non-zero entries to vary across decision rules improved prediction. Motivated by their findings, we specify a hierarchical spike-and-slab prior for $\phi$, which encourages sparsity and also allows the number of non-zero entries of $\phi$ to vary adaptively with the data.

Formally, we introduce a parameter $\theta \in [0, 1]$ that controls the overall sparsity of $\phi$. Conditionally on $\theta$, we draw $p$ binary indicators $\gamma_1, \ldots, \gamma_{p_{\mathrm{cont}}}|\theta \overset{\mathrm{i.i.d.}}{\sim} \mathrm{Bernoulli}(\theta)$. Then, for each $j = 1, \ldots, p_{\mathrm{cont}}$, we draw $\phi_j \sim \mathcal{N}(0, 1)$ if $\gamma_j = 1$ and $\phi_j = 0$ otherwise. Finally, we re-scale $\phi$ to have unit norm. By specifying a further prior on $\theta$, we allow obliqueBART to learn an appropriate level of sparsity of $\phi$ from the data. For simplicity, we specify a conjugate $\mathrm{Beta}(a_\theta, b_\theta)$ prior with fixed $a_\theta, b_\theta > 0$.

Once we draw $\phi$, we draw $c$ uniformly from the range of valid values of $\phi^\top \boldsymbol{x}_{\text{cont}}$ available at nx. This range is determined by the continuous decision rules used at the ancestors of nx in the tree and can be computed by solving two linear programs maximizing and minimizing $\phi^\top \boldsymbol{x}_{\text{cont}}$ over the polytope corresponding to nx.

ObliqueBART depends on several hyperparmeters: the number of trees $M$, the hyperparameters $a_\theta, b_\theta > 0$ for $\theta$, and $\nu$ and $\lambda$ for the prior on $\sigma^2$ (for regression). Following Chipman et al. (2010), we recommend setting $M = 200$, $\nu = 3$, and tuning $\lambda$ so that there is 90% prior probability on the event that $\sigma^2$ is less than the observed variance of $Y$. For the spike-and-slab prior, we recommend fixing $a_\theta = M$ and $b_\theta$ so that the prior mean of $\theta$ is $2/p_{\text{cont}}$. We have found that obliqueBART's performance is not especially sensitive to these choices; see Appendix C.

## 3.2    Posterior computation

For simplicity, we describe the posterior sampling algorithm used for regression. For binary classification, we use essentially the same algorithm but with an additional data augmentation step, which was first described in Albert & Chib (1993).

To simulate draws from the obliqueBART posterior, we deploy a Metropolis-within-Gibbs sampler that is almost identical to the original BART sampler described in Section 2.2. In each Gibbs sampler iteration, we update each of $\mathcal{E}, \theta$, and $\sigma^2$ conditionally on the other two. Simple conjugate updates are available for $\theta$ and $\sigma^2$. Like Chipman et al. (2010), we update the trees in $\mathcal{E}$ one-at-a-time in two steps: a marginal decision tree update followed by a conditional update for the leaf outputs. We further utilize grow and prune proposals to update each decision tree. The only difference between our obliqueBART sampler and the original BART sampler lies in the generation of grow proposals.

### 3.2.1    Regression tree updates

To describe the update of the $m$-th regression tree $(\mathcal{T}_m, \mathcal{D}_m, \mathcal{M}_m) = (\mathcal{T}, \mathcal{D}, \mathcal{M})$, let $\mathcal{E}^-$ denote the collection of the remaining $M - 1$ regression trees in the ensemble. For every node nx in $\mathcal{T}$, let $\mathcal{I}(\text{nx})$ denote the set of indices of the observations $\boldsymbol{x}_i$ whose decision-following paths visit the node nx. Further let $n_{\text{nx}} = |\mathcal{I}(\text{nx})|$ count the number of observations which visit nx. The full conditional posterior density of the $m$-th regression tree factorizes over the leaves of $\mathcal{T}$:

$$p(\mathcal{T}, \mathcal{D}, \mathcal{M}|\boldsymbol{y}, \mathcal{E}^-, \sigma^2, \theta) \propto p(\boldsymbol{y}|\mathcal{T}, \mathcal{D}, \mathcal{M}, \mathcal{E}^-, \sigma^2, \theta)p(\mathcal{M}|\mathcal{T}, \mathcal{D}, \mathcal{E}^-, \sigma^2, \theta)p(\mathcal{T}, \mathcal{D}, |\mathcal{E}^-, \sigma^2, \theta)$$

$$\propto p(\mathcal{T}, \mathcal{D}|\theta) \times \prod_{\substack{\text{leaves} \\ \ell}} \left[ \tau^{-1} \exp \left\{ -\frac{1}{2} \left[ \sigma^{-2} \sum_{i \in \mathcal{I}(\ell)} (r_i - \mu_\ell)^2 + \tau^{-2}\mu_\ell^2 \right] \right\} \right], \quad (1)$$

where $r_i$ is the i-th *partial residual* based on the trees in $\mathcal{E}^-$ given by $r_i = y_i - \sum_{m' \neq m} g(\boldsymbol{x}_i; \mathcal{T}_{m'}, \mathcal{D}_{m'}, \mathcal{M}_{m'})$.

By integrating out the $\mu_\ell$'s from Equation (1), we compute

$$p(\mathcal{T}, \mathcal{D}|\boldsymbol{y}, \mathcal{E}^-, \sigma^2, \theta) \propto p(\mathcal{T}, \mathcal{D}|\theta) \times \prod_{\substack{\text{leaves} \\ \ell}} \left[ \tau^{-1} P_\ell^{-\frac{1}{2}} \exp \left\{ \frac{\Theta_\ell^2}{2P_\ell} \right\} \right], \quad (2)$$

where, for any node nx in $\mathcal{T}$, $P_{\text{nx}} = n_{\text{nx}}\sigma^{-2} + \tau^{-2}$ and $\Theta_{\text{nx}} = \sigma^{-2} \sum_{i \in \mathcal{I}(\text{nx})} r_i$. We can further compute

$$p(\mathcal{M}|\mathcal{T}, \mathcal{D}, \mathcal{E}^-, \boldsymbol{y}, \sigma^2) \propto \prod_{\substack{\text{leaves} \\ \ell}} P_\ell^{\frac{1}{2}} \exp \left\{ -\frac{P_\ell(\mu_\ell - P_\ell^{-1}\Theta_\ell)^2}{2} \right\} \quad (3)$$

We see immediately from Equation (3) that, given the decision tree structure, the leaf outputs are conditionally independent and normally distributed. Specifically, in leaf $\ell$, we have $\mu_\ell \sim \mathcal{N}\left(P_\ell^{-1}\Theta_\ell, P_\ell^{-1}\right)$ conditionally on $\mathcal{T}, \boldsymbol{y}, \mathcal{E}^{(-)}$, and $\sigma^2$. This means that when $n_\ell$ is large, $\mu_\ell$'s conditional posterior is sharply concentrated near the average of the *partial residuals* in the leaf. This is in marked contrast to CART, RF, and ERT where the leaf outputs are exactly the average of the outcomes in the leaf.

**Efficient decision tree updates.** It remains to describe how to sample a new decision tree from the distribution in Equation (2). Although its density is available in closed form, this distribution does not readily admit an exact sampling procedure. Instead, assuming that currently $(\mathcal{T}_m, \mathcal{D}_m) = (\mathcal{T}, \mathcal{D})$, we use a single Metropolis-Hastings (MH) step in which we first propose a random perturbation $(\mathcal{T}^\star, \mathcal{D}^\star)$ of the $(\mathcal{T}, \mathcal{D})$ and then set $(\mathcal{T}_m, \mathcal{D}_m) = (\mathcal{T}^\star, \mathcal{D}^\star)$ with probability

$$\alpha(\mathcal{T}, \mathcal{D} \to \mathcal{T}^\star, \mathcal{D}^\star) = \min \left\{ 1, \frac{q(\mathcal{T}, \mathcal{D} | \mathcal{T}^\star, \mathcal{D}^\star)}{q(\mathcal{T}^\star, \mathcal{D}^\star | \mathcal{T}, \mathcal{D})} \times \frac{p(\mathcal{T}^\star, \mathcal{D}^\star | \boldsymbol{y}, \mathcal{E}^-, \sigma^2, \theta)}{p(\mathcal{T}, \mathcal{D} | \boldsymbol{y}, \mathcal{E}^-, \sigma^2, \theta)} \right\}, \tag{4}$$

where $q(\cdot | \cdot)$ is a to-be-specified proposal kernel.

In obliqueBART, we use a simple proposal mechanism that randomly grows or prunes the tree, each with probability $1/2$. To generate a grow proposal, we (i) attach two child nodes to a leaf node selected uniformly at random; (ii) draw a new decision rule to associate with the selected node; and (iii) leave the rest of the tree and the other decision rules unchanged. To generate a prune proposal, we (i) delete two leaves with a common parent and their incident edges in $\mathcal{T}$; (ii) delete the decision rule associated with the parent of the deleted leaves; and (iii) leave the rest of the tree and the other decision rules unchanged. See Figure 3 for a cartoon illustration.

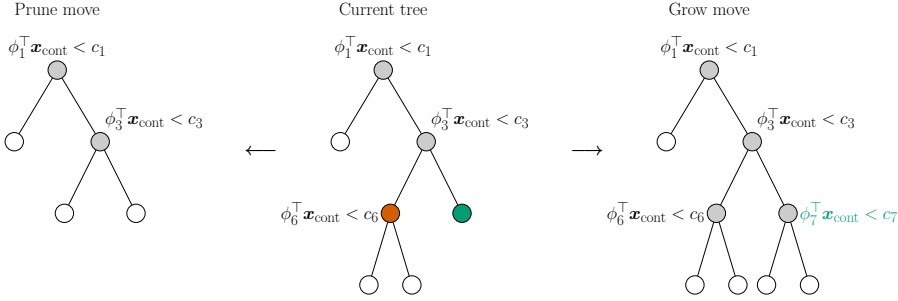

Figure 3: Cartoon illustration of a grow and prune move with oblique, continuous decision rules

The local nature of grow and prune proposals — they change at most two leaves — and the fact that the conditional posterior density of $(\mathcal{T}, \mathcal{D})$ factorizes over leaves (Equation (2)) yields considerable cancelation in the acceptance probability in Equation (4); see Appendix B.

During a grow move, we must draw a decision rule to associate to the newly created decision node from some proposal distribution; in Figure 3, this rule is highlighted in dark green. In principle, we can use any distribution over decision rules for the proposal. For simplicity, we choose to propose decision rules in grow moves from the prior described in Section 3.1. That is, we first randomly decide whether to draw a categorical or continuous decision rule. If we draw a categorical rule, we select the splitting variable and form a random subset of the available levels of that variable. If we decide to draw continuous decision rule, then, conditionally on $\theta$, we draw the entries of $\phi$ independently from the spike-and-slab prior.

Note that there is positive probability that the proposed $\phi$ contains all zeros. When this occurs, we set $c = 1$ so that the decision rule $\{\mathbf{0}^\top \boldsymbol{x} < 1\}$ sends all observations to the left child and none to the right child in the proposed decision tree $(\mathcal{T}^\star, \mathcal{D}^\star)$. Such proposals provide exactly the same fit to the data as the original tree but receive less prior support due to their increased complexity. Consequently, these proposals tend to be rejected in the MH step. When $\phi$ contains at least one non-zero element, we solve two linear programs to compute the maximum and minimum values of $\phi^\top \boldsymbol{x}_{\text{cont}}$ over the linear polytope corresponding to the node `nx`. In our implementation, we solve these programs using the numerical optimization library ALGLIB 4.01.0 (Bochkanov, 2023).

At first glance, it may seem counter-intuitive to propose rules completely at random and independently of the data. One might anticipate that completely random rules would lead to very low MH acceptance probabilities. One might further expect that proposing new rules in a data-dependent fashion, for instance

using the model-based heuristics of Menze et al. (2011), Zhang & Suganthan (2014), or Rainforth & Wood (2015), could lead to larger acceptance probabilities and more efficient MCMC exploration. In fact, drawing overly-informed proposals can result in *slower* MCMC exploration than drawing proposals from the prior.

To develop some intuition for this phenomenon, we note that the MH acceptance probability can be decomposed into the product of two terms (Equation (B1)). The first balances tree complexity against data fit while the other is the ratio between the prior and proposal probabilities of the new rule. Although informed proposals might increase the first term by improving the data fit, they generally make the second term extremely small, deflating the acceptance probability. The resulting Markov chain tends to explore the posterior very slowly. See Deshpande (2024, §2.2 and Appendix B) for more discussion about this phenomenon.

To summarize, in each Gibbs sampler iteration, while keeping $\sigma^2$ and $\theta$ fixed, we first sweep over the trees in $\mathcal{E}$, updating each one conditionally on all the others. Each regression tree update involves (i) sampling a new decision tree from Equation (2) with a MH step and grow/prune mechanism and then (ii), conditionally on the new decision tree, drawing leaf $\ell$'s output from a $\mathcal{N}\left(P_\ell^{-1}\Theta_\ell, P_\ell^{-1}\right)$ distribution. When generating grow proposals, we draw new decision rules from the prior described in Section 3.1.

### 3.2.2 Updating $\theta$ and $\sigma^2$

Once we update each tree in $\mathcal{E}$, we draw $\sigma^2$ and $\theta$ from their full conditional posterior distributions, which are both conjugate. We compute

$$\sigma^2|\boldsymbol{y}, \mathcal{E}, \theta \sim \text{Inv. Gamma}\left(\frac{\nu+n}{2}, \frac{\nu\lambda}{2} + \frac{1}{2}\sum_{i=1}^{n}\left(y_i - \sum_{m=1}^{M} g(\boldsymbol{x}_i; \mathcal{T}_m, \mathcal{D}_m, \mathcal{M}_m)\right)^2\right)$$

$$\theta|\boldsymbol{y}, \mathcal{E}, \sigma^2 \sim \text{Beta}\left(a_\theta + n_\phi, b_\theta + z_\phi\right),$$

where $n_\phi$ (resp. $z_\phi$) counts the number of non-zero (resp. zero) entries in the $\phi$'s appearing in $\mathcal{E}$.

An R package implementing obliqueBART is available at https://github.com/paulhnguyen/obliqueBART.

## 4 Experiments

We compared the performance of obliqueBART to other tree models using several synthetic and benchmark datasets. In Section 4.1, we use the synthetic data from Figure 2 to compare obliqueBART to axis-aligned BART with the original features and several randomly rotated versions of the features. Then, in Section 4.2, we compare obliqueBART to other axis-aligned and oblique tree methods, as well as axis-aligned tree ensemble methods fit with randomly pre-rotated features on benchmark datasets for both regression and classification problems. Generally speaking, obliqueBART run with its default hyperparameter values performed very well on regression tasks and was competitive with the other tuned methods on classification tasks. We further found that randomly pre-rotating features before fitting axis-aligned RF, ERT, or GBT models tended to yield worse results than simply fitting obliqueBART.

We performed all of our experiments on a shared high-throughput computing cluster. For obliqueBART and BART, we compute posterior means of $f(\boldsymbol{x})$ (for regression) and $\mathbb{P}(y = 1|\boldsymbol{x})$ (for classification) based on 1000 samples obtained by simulating a single Markov chain for 2000 iterations and discarding the first 1000 as "burn-in." We fit BART, RF, ERT, and XGBoost models using implementations available in the R packages **BART** (Sparapani et al., 2021), **randomForest** (Liaw & Wiener, 2002), **ranger** (Wright & Ziegler, 2017), and **xgboost** (Chen et al., 2024). We also compared obliqueBART's performance to rotated versions of RF, ERT, XGB, and BART with $R = 200$ random rotations. Additionally, we fit oblique tree ensembles from the **ODRF** (Zhan et al., 2022) and **aorsf** (Jaeger et al., 2022) packages on both the classification and regression settings. We also compared obliqueBART to oblique tree-based classifiers implemented in the **RPEnsemble** (Cannings & Samworth, 2017) and **rotationForest** (Ballings & Van den Poel, 2017) packages.

## 4.1 Synthetic data experiments

We compared obliqueBART to axis-aligned BART with data from the two simple functions in Figure 2. Since obliqueBART partitions the feature space along randomly selected directions, we additionally compared it with a hybrid procedure that first computes $R$ random rotations of the feature space and then trains an axis-aligned BART model using the rotated features. This random rotation BART model is a analogous to the random rotation ensemble methods studied in Blaser & Fryzlewicz (2016), but it does not screen or tune the random rotations.

For these experiments, we sampled $n$ covariate vectors $\boldsymbol{x}_i \sim \text{Uniform}([0,1]^2)$ and generated $y_i \sim \mathcal{N}\left(f(\boldsymbol{x}_i; \theta, \Delta), 1\right)$, where the function $f(\boldsymbol{x}; \theta, \Delta)$ takes on two values $\pm\Delta$ and depends on a parameter $\theta$. We considered two such functions, the rotated axes function (Figure 2a) and the sinusoid function (Figure 2d).

**Rotated axes**. Given any $\theta \in [0, \pi/4]$, let $\boldsymbol{u}(\boldsymbol{x})$ be the point obtained by rotating $\boldsymbol{x}$ $\theta$ radians counterclockwise around the origin. The value of rotated axes function is determined by the quadrant in which $\boldsymbol{u}$ lies. Specifically, for rotation angle $\theta \in \{0, \pi/36, \dots, \pi/4\}$ and jump $\Delta \in \{0.5, 1, 2, 4\}$, we set $f(\boldsymbol{x}; \theta, \Delta) = \Delta \times (2 \times \mathbb{1}(u_1 u_2 > 0) - 1)$.

**Sinusoid**. The value of $f(\boldsymbol{x}; \theta, \Delta)$ depends on which side of a particular sinusoid the point $\boldsymbol{x}$ lies. Specifically, for a given amplitude $\theta \in \{0, 0.1, \dots, 1\}$ and jump $\Delta \in \{0.5, 1, 2, 4\}$, we set $f(\boldsymbol{x}; \theta, \Delta) = \Delta \times (2 \times \mathbb{1}(x_2 > \theta \sin(10x_1)) - 1)$.

For each combination of $n \in \{100, 1000, 10000\}$, function $f$, jump $\Delta$, and parameter $\theta$, we generated 20 datasets of size $n$. Figure 4 shows how axis-aligned BART and random rotation BART's out-of-sample RMSE (averaged over 20 replications) relative to obliqueBART changes as the decision boundaries becomes less and less axis-aligned (i.e., as $\theta$ increases) with $\Delta = 4$.

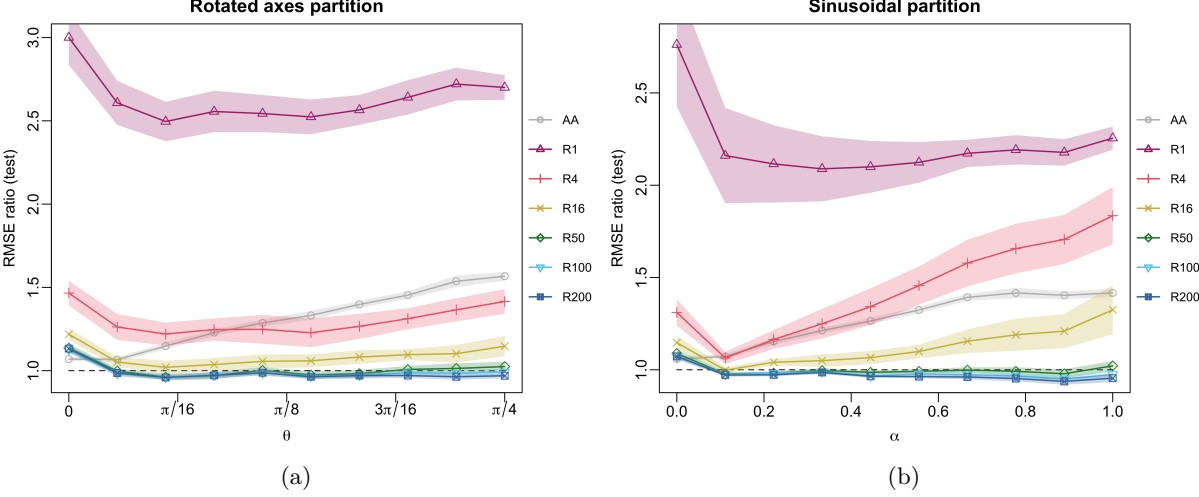

(a)                                     (b)

Figure 4: Performance of axis-aligned BART (AA) and axis-aligned BART with random rotations relative to obliqueBART in terms of out-of-sample predictive error in the (a) rotated axes partition and (b) sinusoidal partition, as pictured in Figure 2. This figure depicts the experiments run with $n = 1000$ and $\Delta = 4$. Shaded regions depict 95% confidence intervals for the ratio of each model's out-of-sample RMSE and obliqueBART's out-of-sample RMSE are shaded (i.e., the out-of-sample error relative to obliqueBART).

As we might expect, obliqueBART substantially outperformed axis-aligned BART on the rotated axes problem. Interestingly, we see that obliqueBART even out-performs axis-aligned BART when $\theta = 0$ and the true decision boundary is axis-aligned. When $\theta = 0$, the obliqueBART ensemble used, on average, axis-aligned rules 70.2% of the time compared to 52.9% of the time when $\theta = \pi/4$. This suggests that obliqueBART, through the hierarchical spike-and-slab prior for $\phi$, can adaptively adjust the number of axis-aligned or

oblique rules. ObliqueBART further outperformed axis-aligned BART on the sinusoid problem, even though the decision boundary was highly non-linear (see Figure 4b).

Although obliqueBART also outperformed random rotation BART with a small number of random rotations, the gap between the methods diminished as the number of rotations $R$ increased. For these data, random rotation BART run with at least 50 rotations performed as well as — and, at times, slightly outperformed — obliqueBART. As we discuss in Section 4.2, matching obliqueBART's performance with a random rotation ensemble is much harder when the number of continuous predictors $p_{cont}$ is large.

## 4.2 Results on benchmark datasets

The regression datasets contain between 96 and 53,940 observations and between 4 and 31 predictors and the classification datasets contain between 61 and 4601 observations and between 4 and 72 predictors. In our experiments, continuous variables were scaled to $[-1, 1]$ and observations with missing values were dropped before we fit any models to the data. We obtained most of these datasets from UCI Machine Learning Repository (https://archive.ics.uci.edu); the *Journal of Applied Econometrics* data archive (http://qed.econ.queensu.ca/jae/); and from several R packages. See Table A1 for the dimensions of and links to these datasets. For each competing method, we tuned hyperparameters using 5-fold cross-validation on each training dataset. Table A2 shows the grids of values considered for each method's hyperparameters. These grids mirror those used in Chipman et al. (2010, Table 2).

We created 20 random 75%-25% training-testing splits of each dataset. For regression tasks, we computed each methods' standardized out-of-sample mean square error (SMSE), which is defined as the ratio between the the mean square errors of a model, $n_{test}^{-1} \sum_{i=1}^{n_{test}} (Y_{test,i} - \hat{Y}_i)^2$ and the mean of the training data $n_{test}^{-1} \sum_{i=1}^{n_{test}} (Y_{test,i} - \overline{Y}_{train})^2$. For classification problems, we formed predictions by truncating the outputted class probability at 50% and computed out-of-sample accuracy. We performed one-sided paired t-tests to determine whether obliqueBART's error were significantly less than the competing methods' errors.

### 4.2.1 Comparison to axis-aligned methods

Figure 5 compares obliqueBART's SMSE and accuracy to those of the axis-aligned methods across every fold and dataset. We defer dataset-by-dataset tabulations of these error metrics to Tables A3 and A4 in Appendix A.

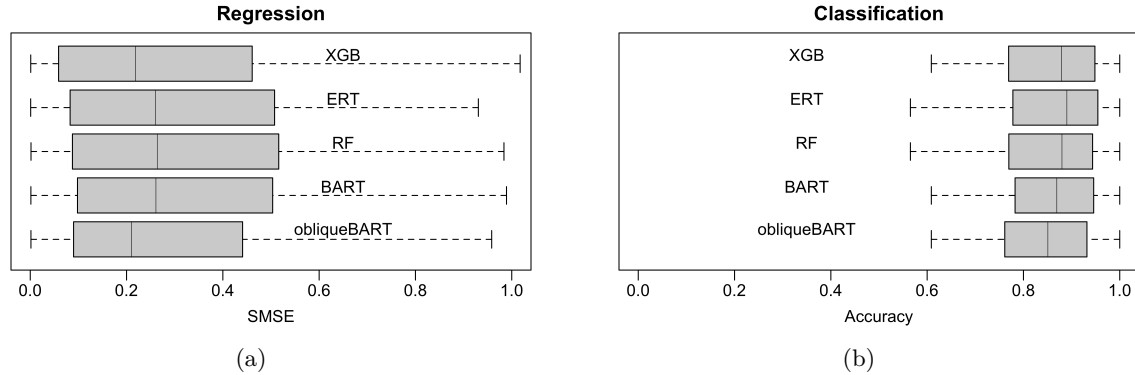

Figure 5: ObliqueBART's SMSE (a) and accuracy (b) across all splits and datasets, compared to XGB, ERT, RF, and BART. Models with lower SMSE's and higher accuracies are preferred.

Although no single method performed the best across all regression datasets, obliqueBART had the smallest overall mean SMSE (0.297). For comparison, the mean SMSEs for the tuned axis-aligned methods were 0.316 (BART), 0.314 (RF), 0.309 (ERT), and 0.299 (XGB). The median SMSEs were respectively 0.210 (obliqueBART), 0.261 (BART), 0.264 (RF), 0.260 (ERT), and 0.219 (XGB). ObliqueBART's SMSE was also statistically significantly lower (at the 5% level) than BART's on nine datasets, RF's on eight datasets,

ERT's on six datasets, and XGB's on five datasets. It is notable that the "off-the-shelf" performance of obliqueBART on regression tasks was somewhat better than the performance of its tuned competitors.

ObliqueBART had the lowest classification accuracy (0.846) averaged across all folds and datasets while ERT had the highest (0.867). ERT had the largest accuracy on eleven datasets while obliqueBART was the best-performing method for four datasets. Generally speaking, however, obliqueBART was still competitive with the other methods for classification. For example, the difference in accuracy between obliqueBART and ERT, the best method, was less than 2% in 13 of the 22 datasets. Similarly, the difference in accuracy between obliqueBART and BART was less than 1% in 13 of the datasets.

Because BART and obliqueBART generate many samples of the regression tree ensemble, we would expect them to be slower than RF, ERT, and XGB, which only generate a single ensemble. While BART and obliqueBART typically take longer to compute compared to non-Bayesian tree ensembles, their ability to quantify uncertainty often make the increased computation time worthwhile. Further, because obliqueBART involves solving two linear programs when proposing new decision rules, we would expect it to be slower than BART. This was generally the case in our experiments: for most datasets, obliqueBART was about twice as slow as BART and slower than RF, ERT, and XGB. Surprisingly, obliqueBART was much faster on the `diamonds` dataset than BART.

### 4.2.2 Comparison with oblique ensembles

Next, we compared obliqueBART's performance on the benchmark datasets with available oblique tree ensemble implementations in R. For each competitor method, we tuned hyperparameters based on 5-fold cross validation within each training set.

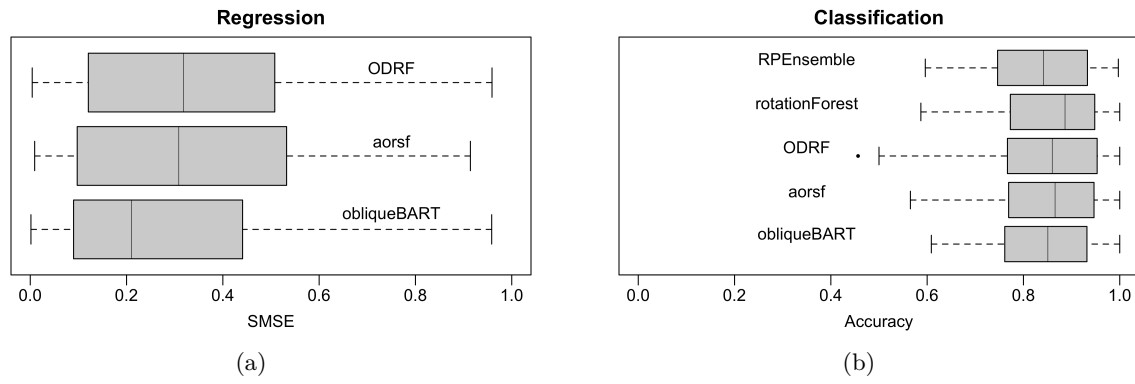

Figure 6: ObliqueBART's SMSE (a) and accuracy (b) across all splits and datasets, compared to ODRF, aorsf, RPEnsemble, and rotationForest. Models with lower SMSE's and higher accuracies are preferred.

ObliqueBART still performs quite well in the regression setting compared to other oblique ensembles. ObliqueBART has the lowest mean and median SMSE across all datasets and splits at 0.297 and 0.210, respectively; ODRF has mean and median SMSE at 0.339 and 0.318, and aorsf has mean and median SMSE 0.343 and 0.309. For classification, obliqueBART is average compared to the other oblique ensembles, and no specific method stands out from the rest, generally.

One important distinction in these results is the reliability of each implementation. ObliqueBART runs for all 40 benchmark datasets, for each training and testing split. In our experiments, we found that some other methods often failed, either because they hit memory limits or encountered values of a categorical level in testing that were not seen in training. Tables A5 and A6 contain the mean SMSE and accuracy for each individual dataset.

### 4.2.3 Comparison with random rotation ensembles

Next, we compared obliqueBART's performance to randomly rotated versions of BART, RF, ERT, and XGB with $R \in \{1, 4, 16, 50, 100, 200\}$ random rotations. We report dataset-by-dataset SMSEs and accuracies in

Tables A7 and A8. We additionally determined how many random rotations were needed for each of these methods to match obliqueBART's performance (Tables A9 and A10). Figure 7 shows the SMSEs and accuracies of obliqueBART and the random rotation ensembles with $R = 200$ for all datasets and folds.

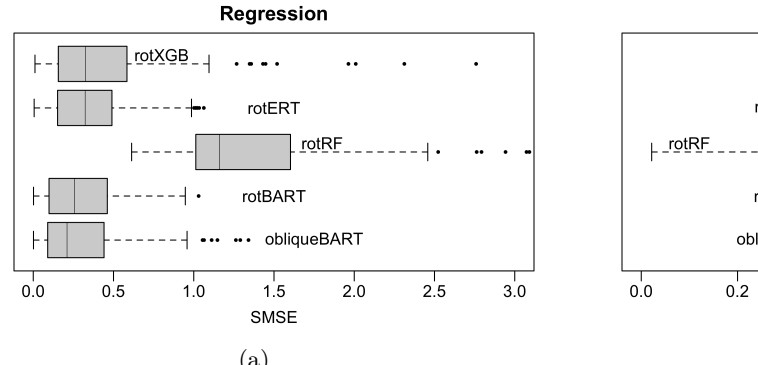
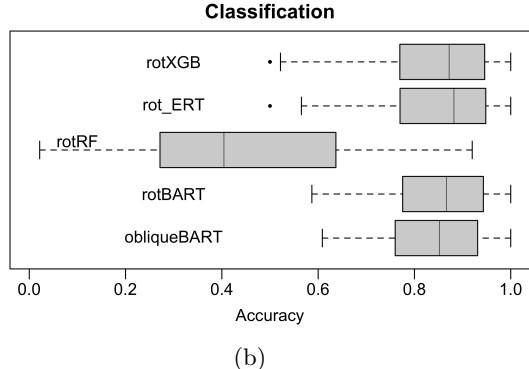

| (a) | (b) |

Figure 7: ObliqueBART's SMSE and accuracy (resp. left and right) across all splits and datasets, compared to XGB, ERT, RF, and BART with 200 random rotations. Models with lower SMSE's and higher accuracies are preferred.

Somewhat surprisingly, the randomly rotated random forests (rotRF) performed markedly worse than the other methods. For regression, obliqueBART and the rotated version of BART (rotBART) had the smallest average SMSEs (0.297 and 0.296, resp.). ObliqueBART was also competitive with the rotERT and rotXGB for classification; its average accuracy was 0.846 while the best performing method, rotERT, had an average accuracy of 0.859. We additionally observed that obliqueBART was, on average, 20 times faster than rotRF and twice as fast as rotBART run with $R = 200$ random rotations.

Interestingly, there was substantial variation in the minimum number of rotations needed for rotBART, rotRF, rotERT, and rotXGB to match obliqueBART's performance. For instance, rotBART was unable to match obliqueBART's performance on 19 datasets even with 200 rotations and required only one rotation for 17 datasets; four rotations for two datasets; 16 rotations for one dataset; and 50 rotations for one dataset. rotXGB, on the other hand, was unable to match obliqueBART's performance on 24 datasets using 200 rotations and required one rotation for nine datasets; four rotations for four datasets; 16 rotations for one dataset; 50 rotations for one dataset; and 100 rotations for one dataset. These results suggest that running obliqueBART is often more effective and faster than tuning the number of random pre-rotations used to train a random rotation ensemble.

## 5 Discussion

We introduced obliqueBART, which extends the expressivity of the BART model by building regression trees that partition the predictor space based on random hyperplanes. Unlike oblique versions of CART or RF, obliqueBART does not search for optimal decision rules and instead grows trees by randomly accepting (via a Metropolis-Hastings step) completely *random* decision rules. Although obliqueBART does not uniformly outperform BART across the 40 benchmark datasets we considered, its performance is generally not significantly worse than BART's and can sometimes be substantially better. On this view, we would not advocate for a wholesale replacement axis-aligned BART in favor of our obliqueBART implementation. It is possible, for instance, that alternative priors for $\phi$ may yield somewhat larger improvements.

While we have focused primarily on the tabular data setting, we anticipate that our obliqueBART implementation can be fruitfully extended to accommodate structured input like images. Li et al. (2023a) demonstrated that oblique tree ensembles can close the performance gap between non-neural network methods and convolutional deep networks for image classification. Their manifold oblique random forests (MORF) procedure recursively partitions images based on the average pixel value within random rectangular sub-regions of an image. Developing a BART analog of MORF would involve modifying the decision rule prior to ensure that

the non-zero elements of $\phi$ correspond to a connected sub-region of an image. We leave this and similar extensions for other types of structured inputs (e.g., tensors, functional data) to future work.

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

# A  Additional benchmarking details and results

Table A1 lists the dimensions and sources of each benchmark dataset. Most datasets are from the UCI data repository, though some are from individual R packages (bolded), the Journal of Applied Econometrics data archive, and the CMU Statlib data repository.

Table A1: Dimensions and hyperlinked sources of benchmark datasets

| Regression Data | | | | Classification Data | | | |
|---|---|---|---|---|---|---|---|
| Data (Source) | $n$ | $p$ | $p_{cont}$ | Data | $n$ | $p$ | $p_{cont}$ |
| abalone (UCI) | 4177 | 8 | 7 | banknote(UCI) | 1372 | 4 | 4 |
| ais (**DAAG**) | 202 | 12 | 10 | blood transfusion (UCI) | 748 | 4 | 4 |
| ammenity (JAE) | 3044 | 25 | 20 | breast cancer diag (UCI) | 569 | 30 | 30 |
| attend (JSE) | 838 | 9 | 6 | breast cancer (UCI) | 683 | 9 | 9 |
| baseball(**ISLR**) | 263 | 19 | 16 | breast cancer prog. (UCI) | 194 | 32 | 32 |
| basketball (SMIS) | 96 | 4 | 3 | climate crashes (UCI) | 540 | 18 | 18 |
| boston (**MASS**) | 506 | 4 | 3 | connectionist sonar (UCI) | 208 | 60 | 60 |
| budget (JAE) | 1729 | 10 | 10 | credit approval (UCI) | 653 | 15 | 6 |
| cane (OzDASL) | 3775 | 31 | 25 | echocardiogram (UCI) | 579 | 10 | 9 |
| cpu (UCI) | 209 | 7 | 6 | fertility (UCI) | 100 | 9 | 3 |
| diabetes (**lars**) | 442 | 10 | 9 | german credit (UCI) | 1000 | 20 | 7 |
| diamonds (**ggplot2**) | 53940 | 9 | 6 | heptatitis (UCI) | 80 | 19 | 7 |
| edu (JAE) | 2338 | 6 | 5 | ILPD (UCI) | 579 | 10 | 9 |
| labor (**Ecdat**) | 5320 | 6 | 4 | ionosphere (UCI) | 351 | 34 | 34 |
| mpg (UCI) | 392 | 8 | 7 | ozone1 (UCI) | 1848 | 72 | 72 |
| rice (JAE) | 1026 | 18 | 14 | ozone8 (UCI) | 1847 | 72 | 72 |
| servo (UCI) | 167 | 4 | 2 | parkinsons (UCI) | 195 | 22 | 22 |
| strikes (Statlib) | 625 | 6 | 5 | planning relax (UCI) | 182 | 12 | 12 |
| | | | | qsar bio. (UCI) | 1055 | 41 | 38 |
| | | | | seismic bumps (UCI) | 2584 | 18 | 14 |
| | | | | spambase (UCI) | 4601 | 57 | 57 |
| | | | | spectf heart (UCI) | 267 | 44 | 44 |

Table A2 shows the values considered for each competing method's operational parameters. We used functions included in the **RPEnsemble** package to choose the best values for its hyperparameters.

Table A2: Operational parameters for competing methods

| Method | Parameter | Values considered |
|---|---|---|
| Random Forest | # of trees | 10, 50, 100, 200 |
| | % of variables sampled to grow node | 10, 33, $\frac{100}{\sqrt{p}}$, 50 |
| Gradient Boosting | # of trees | 50, 100, 200 |
| | Shrinkage | .01, .05, .1, .25 |
| | Max depth for each tree | 1, 2, 3, 4 |
| Extremely Randomized Trees | # of trees | 10, 50, 100, 200 |
| | % of variables sampled to grow node | 10, 33, $\frac{100}{\sqrt{p}}$, 50 |
| | # of random splits to consider | 1, 2 |
| | | |
| Rotation Forest | # of trees | 10, 50, 100, 200 |
| | % of variables sampled to grow node | 10, 33, $\frac{100}{\sqrt{p}}$ |
| ODRF | # of trees | 10, 50, 100, 200 |
| AORSF | # of trees | 10, 50, 100, 200 |
| | % of variables sampled to grow node | 10, 33, $\frac{100}{\sqrt{p}}$, 50 |
| | # of cut-points assessed | 1, 2, 5 |

Tables A3 and A4, respectively, show the out-of-sample standardized mean square and accuracy of oblique-BART, RF, ERT, BART, and XGB for each benchmark dataset, averaged over 20 training-testing splits. We marked entries in these tables with asterisks whenever obliqueBART achieved statistically significantly lower SMSE or higher accuracy than the competing method. Throughout, we assessed significance using a paired t-test and a 5% threshold. We report analogous results from comparison obliqueBART to rotated versions of RF, ERT, BART, and XGB run with 200 random rotations in Tables A7 and A8.

Table A3: Standardized mean square errors on regression benchmark datasets, averaged across 20 training-testing splits, for obliqueBART and axis-aligned methods. Best performing method is bolded and errors that are statistically significantly larger than obliqueBART's are marked with an asterisk.

| data | obliqueBART | BART | ERT | RF | XGB |
|------|-------------|------|-----|-----|-----|
| abalone | **0.44** | *0.449 | *0.45 | *0.449 | *0.454 |
| ais | 0.122 | *0.133 | **0.108** | 0.112 | 0.131 |
| amenity | 0.283 | 0.278 | 0.279 | 0.282 | **0.274** |
| attend | **0.246** | *0.302 | *0.412 | *0.428 | 0.246 |
| baseball | **0.39** | *0.399 | *0.439 | *0.419 | *0.461 |
| basketball | 0.762 | 0.669 | **0.656** | 0.708 | 0.753 |
| boston | 0.778 | 0.6 | **0.587** | 0.62 | 0.639 |
| budget | 0.003 | 0.003 | **0.002** | 0.004 | 0.003 |
| cane | 0.183 | *0.193 | 0.18 | 0.182 | **0.173** |
| cpu | **0.139** | *0.272 | 0.149 | *0.178 | 0.153 |
| diabetes | **0.495** | *0.5 | *0.526 | *0.546 | *0.537 |
| diamonds | 0.02 | 0.019 | **0.018** | 0.018 | 0.02 |
| edu | 0.02 | 0.02 | 0.02 | *0.02 | **0.02** |
| labor | **0.221** | *0.693 | *0.604 | *0.535 | *0.349 |
| mpg | 0.135 | **0.125** | 0.127 | 0.135 | *0.144 |
| rice | 0.015 | *0.027 | *0.019 | *0.027 | **0.012** |
| servo | 0.167 | 0.149 | 0.169 | 0.152 | **0.073** |
| strikes | 0.926 | 0.863 | **0.816** | 0.836 | 0.94 |

Table A4: Accuracies on classification benchmark datasets, averaged across 20 training-testing splits for obliqueBART and axis-aligned methods. Best performing method is bolded and errors that are statistically significantly larger than obliqueBART's are marked with an asterisk.

| data | obliqueBART | BART | ERT | RF | XGB |
|---|---|---|---|---|---|
| banknote | 0.997 | 0.997 | **0.998** | *0.991 | 0.995 |
| blood transfusion | 0.769 | **0.795** | 0.79 | 0.767 | 0.783 |
| breast cancer diag. | 0.922 | **0.966** | 0.965 | 0.958 | 0.965 |
| breast cancer | 0.975 | *0.973 | **0.976** | 0.972 | *0.965 |
| breast cancer prog. | 0.758 | **0.76** | *0.744 | *0.745 | 0.751 |
| climate crashes | 0.917 | 0.944 | 0.944 | 0.918 | **0.948** |
| connectionist sonar | 0.802 | 0.824 | **0.861** | 0.808 | 0.831 |
| credit approval | 0.866 | 0.87 | 0.87 | **0.871** | 0.865 |
| echocardiogram | 0.714 | 0.712 | **0.719** | *0.699 | *0.694 |
| fertility | **0.86** | **0.86** | 0.852 | *0.836 | 0.852 |
| hepatitis | 0.84 | 0.865 | **0.87** | 0.86 | 0.85 |
| ILPD | 0.714 | 0.71 | **0.719** | *0.699 | *0.694 |
| ionosphere | 0.902 | 0.926 | **0.941** | 0.92 | 0.923 |
| ozone 1 | **0.97** | 0.97 | 0.969 | 0.97 | 0.969 |
| ozone 8 | 0.933 | 0.938 | **0.941** | 0.938 | 0.939 |
| parkinsons | 0.865 | 0.861 | **0.913** | 0.89 | 0.892 |
| planning relax | **0.717** | *0.708 | *0.695 | *0.677 | 0.71 |
| qsar biodegradable | 0.823 | 0.845 | **0.866** | 0.86 | 0.863 |
| seismic bumps | **0.935** | 0.935 | 0.935 | *0.931 | *0.934 |
| spambase | 0.771 | 0.932 | **0.956** | 0.948 | 0.952 |
| spectf heart | 0.799 | **0.813** | 0.795 | 0.806 | 0.787 |
| statlog german cred. | 0.751 | **0.76** | 0.759 | 0.757 | 0.749 |

Table A5: SMSE for obliqueBART and other oblique ensemble methods on regression benchmark datasets, averaged across 20 training-testing splits. The best performing method is bolded; model and data combinations that were not run succesfully across all 20 train/test splits are marked with parentheses.

| data | obliqueBART | ODRF | aorsf |
|---|---|---|---|
| abalone | 0.44 | **0.414** | 0.419 |
| ais | 0.122 | 0.121 | **0.114** |
| amenity | **0.283** | 0.319 | 0.324 |
| attend | **0.246** | 0.446 | 0.491 |
| baseball | **0.39** | 0.427 | 0.477 |
| basketball | 0.762 | **0.636** | 0.659 |
| boston | 0.778 | **0.585** | 0.67 |
| budget | **0.003** | 0.01 | 0.013 |
| cane | **0.183** | 0.186 | 0.259 |
| cpu | **0.139** | 0.282 | 0.28 |
| diabetes | 0.495 | 0.494 | **0.493** |
| diamonds | 0.02 | NA | **0.018** |
| edu | 0.02 | **0.019** | 0.021 |
| labor | **0.221** | 0.674 | 0.787 |
| mpg | 0.135 | **0.132** | 0.135 |
| rice | **0.015** | 0.03 | 0.038 |
| servo | **0.167** | 0.192 | 0.218 |
| strikes | 0.926 | **0.795** | (0.801) |

Table A6: Accuracies for obliqueBART and other oblique ensemble methods on classification benchmark datasets, averaged across 20 training-testing splits. The best performing method is bolded; model and data combinations that were not run succesfully across all 20 train/test splits are marked with parentheses.

| data | obliqueBART | ODRF | RPEnsemble | aorsf | rotationForest |
|---|---|---|---|---|---|
| banknote | 0.997 | 0.994 | 0.985 | **0.998** | 0.993 |
| blood transfusion | 0.769 | **0.797** | 0.772 | 0.793 | 0.789 |
| breast cancer diag. | 0.922 | 0.975 | 0.943 | **0.977** | 0.954 |
| breast cancer | 0.975 | 0.973 | 0.972 | **0.976** | 0.975 |
| breast cancer prog. | **0.758** | 0.727 | 0.756 | 0.756 | (0.748) |
| climate crashes | 0.917 | **0.956** | 0.917 | 0.92 | 0.934 |
| connectionist sonar | **0.802** | 0.8 | 0.757 | 0.788 | (0.783) |
| credit approval | 0.866 | **0.872** | 0.871 | 0.867 | (0.871) |
| echocardiogram | **0.714** | 0.703 | 0.714 | 0.707 | 0.706 |
| fertility | **0.86** | 0.842 | 0.85 | **0.86** | NA |
| hepatitis | 0.84 | 0.835 | 0.845 | 0.84 | (**0.885**) |
| ILPD | **0.714** | 0.703 | 0.714 | 0.707 | 0.706 |
| ionosphere | 0.902 | NA | 0.895 | NA | (**0.946**) |
| ozone 1 | **0.97** | 0.968 | **0.97** | **0.97** | 0.97 |
| ozone 8 | 0.933 | 0.94 | 0.933 | 0.939 | **0.942** |
| parkinsons | 0.865 | 0.878 | 0.855 | 0.87 | (**0.881**) |
| planning relax | **0.717** | 0.645 | 0.716 | 0.711 | 0.696 |
| qsar biodegradable | 0.823 | **0.866** | 0.709 | 0.864 | 0.858 |
| seismic bumps | **0.935** | NA | **0.935** | NA | 0.934 |
| spambase | 0.771 | **0.951** | (0.724) | 0.947 | (0.925) |
| spectf heart | 0.799 | 0.802 | 0.799 | **0.808** | (0.806) |
| statlog german cred. | 0.751 | 0.76 | 0.746 | (0.759) | **0.763** |

Table A7: Standardized mean square errors on regression benchmark datasets, averaged across 20 training-testing splits, for obliqueBART and rotated versions of axis-aligned methods with 200 random rotations. Best performing method is bolded and errors that are statistically significantly larger than obliqueBART's are marked with an asterisk. NA indicates that the method could not be run with 200 random rotations.

| data | obliqueBART | rotBART | rotERT | rotRF | rotXGB |
|---|---|---|---|---|---|
| abalone | **0.44** | 0.448* | 0.45* | 1.08* | 0.47* |
| ais | **0.122** | 0.129 | 0.151* | 1.33* | 0.167* |
| amenity | 0.283 | **0.281** | 0.349* | 1.07* | 0.357* |
| attend | **0.246** | 0.297* | 0.304* | 1.06* | 0.305* |
| baseball | **0.39** | 0.406* | 0.446* | 1.07* | 0.544* |
| basketball | 0.762 | 0.691 | **0.68** | 1.06* | 0.767* |
| boston | 0.778 | **0.577** | 0.833 | 22.6* | 0.853 |
| budget | **0.003** | 0.003 | 0.010* | 1.200* | 0.019* |
| cane | **0.183** | 0.191 | 0.315* | 1.580* | 0.294* |
| cpu | **0.139** | 0.313* | 0.201* | 1.480* | 0.249* |
| diabetes | **0.495** | 0.499 | 0.520* | 0.967* | 0.606* |
| diamonds | 0.020 | **0.020** | NA | NA | NA |
| edu | **0.020** | 0.020 | 0.022* | 1.150* | 0.024* |
| labor | **0.221** | NA | NA | NA | NA |
| mpg | **0.135** | 0.138 | 0.14 | 1.09* | 0.16* |
| rice | **0.015** | 0.026* | 0.204* | 1.670* | 0.234* |
| servo | **0.167** | 0.196* | 0.25* | 2.24* | 0.225* |
| strikes | 0.926 | **0.805** | 0.915 | 36.2* | 1.31 |

Table A8: Accuracies on classification datasets, averaged across 20 training-testing splits, for obliqueBART and rotated versions of axis-aligned methods with 200 random rotations. Best performing method is bolded and accuracies that are statistically significantly worse than obliqueBART's are marked with an asterisk.

| data | obliqueBART | rotBART | rotERT | rotRF | rotXGB |
|---|---|---|---|---|---|
| banknote | 0.997 | 0.999 | **1** | *0.451 | 0.998 |
| blood transfusion | 0.769 | **0.793** | *0.734 | *0.748 | 0.789 |
| breast cancer diag. | 0.922 | 0.973 | **0.977** | *0.635 | 0.97 |
| breast cancer | 0.975 | **0.975** | 0.973 | *0.647 | 0.972 |
| breast cancer prog. | 0.758 | **0.759** | 0.751 | *0.303 | *0.733 |
| climate crashes | 0.917 | 0.932 | 0.926 | *0.56 | **0.952** |
| connectionist sonar | 0.802 | 0.801 | **0.849** | *0.474 | 0.824 |
| credit approval | 0.866 | *0.859 | 0.87 | *0.453 | **0.871** |
| echocardiogram | **0.714** | 0.708 | 0.712 | *0.343 | *0.695 |
| fertility | **0.86** | **0.86** | *0.848 | *0.726 | *0.84 |
| hepatitis | 0.84 | **0.858** | **0.858** | *0.725 | 0.842 |
| ILPD | **0.714** | *0.7 | 0.711 | *0.336 | 0.708 |
| ionosphere | 0.902 | *0.881 | **0.93** | *0.361 | 0.892 |
| ozone 1 | **0.97** | *0.97 | *0.968 | *0.0314 | 0.969 |
| ozone 8 | 0.933 | **0.941** | 0.939 | *0.0695 | 0.939 |
| parkinsons | 0.865 | 0.859 | **0.911** | *0.265 | 0.894 |
| planning relax | **0.717** | *0.711 | *0.653 | *0.283 | *0.63 |
| qsar bio. | 0.823 | 0.847 | 0.851 | *0.568 | **0.857** |
| seismic bumps | **0.935** | *0.934 | *0.919 | *0.0648 | *0.929 |
| spambase | 0.771 | 0.936 | **0.951** | *0.413 | 0.945 |
| spectf heart | 0.799 | **0.817** | 0.813 | 0.799 | 0.791 |

Table A9: Number of random rotations of the data for difference between obliqueBART's and rotated method's MSE to be statistically insignificant. Model and data combinations with "-" had MSE's that were larger than obliqueBART, even with 200 random data rotations.

| data | rotBART | rotERT | rotRF | rotXGB |
|---|---|---|---|---|
| abalone | - | - | - | - |
| ais | 1 | - | - | - |
| amenity | 1 | - | - | - |
| attend | - | - | - | - |
| baseball | - | - | - | - |
| basketball | - | - | - | 1 |
| boston | - | 1 | 1 | 1 |
| budget | 1 | - | - | - |
| cane | 50 | - | - | - |
| cpu | - | - | - | - |
| diabetes | 1 | - | - | - |
| diamonds | - | - | - | - |
| edu | 1 | - | - | - |
| labor | - | - | - | - |
| mpg | 1 | 16 | - | - |
| rice | - | - | - | - |
| servo | - | - | - | 100 |
| strikes | - | 4 | 1 | - |

Table A10: Number of random rotations of the data for difference between obliqueBART's and rotated method's accuracies to be statistically insignificant. Model and data combinations with "-" had smaller accuracies than obliqueBART, even with 200 random data rotations.

| data | rotBART | rotERT | rotRF | rotXGB |
|------|---------|--------|-------|--------|
| banknote | 1 | 1 | 1 | 4 |
| blood transfusion | - | - | 4 | - |
| breast cancer diag. | - | - | - | - |
| breast cancer | 1 | 1 | 1 | 4 |
| breast cancer prog. | 1 | 1 | 1 | 1 |
| climate crashes | - | 1 | 4 | - |
| connectionist sonar | 1 | - | - | 1 |
| credit approval | 1 | 1 | 1 | 1 |
| echocardiogram | 1 | 1 | 1 | 16 |
| fertility | 1 | 4 | 4 | 16 |
| hepatitis | - | 16 | 4 | 1 |
| ILPD | 1 | 1 | 1 | 4 |
| ionosphere | - | - | 1 | 50 |
| ozone 1 | 1 | - | - | 1 |
| ozone 8 | - | - | - | - |
| parkinsons | 1 | - | - | 1 |
| planning relax | 16 | - | - | - |
| qsar bio. | - | - | - | - |
| seismic bumps | 1 | - | - | 1 |
| spambase | - | - | - | - |
| spectf heart | 1 | 4 | 4 | 1 |

## B    Metropolis-Hastings acceptance probabilities

**Grow move.** Suppose we are updating the $m$-th decision tree $(\mathcal{T}_m, \mathcal{D}_m) = (\mathcal{T}, \mathcal{D})$. Further suppose that we formed the proposal $(\mathcal{T}^\star, \mathcal{D}^\star)$ by growing $(\mathcal{T}, \mathcal{D})$ from an existing leaf $\mathtt{nx}$ at depth $d(\mathtt{nx})$ and then drawing a new rule $\mathtt{rule}$ to associate with $\mathtt{nx}$ in $(\mathcal{T}^\star, \mathcal{D}^\star)$. Let $q(\mathtt{rule}|\mathcal{T}, \mathcal{D})$ denote the proposal probability of drawing $\mathtt{rule}$ at $\mathtt{nx}$. Additionally, let $\mathtt{nleaf}(\cdot)$ and $\mathtt{nnog}(\cdot)$ count the number of leaf nodes and decision nodes with no grandchildren in a tree. The acceptance probability of a grow move decomposes the product of three terms.

$$
\alpha(\mathcal{T}, \mathcal{D} \to \mathcal{T}^\star, \mathcal{D}^\star) = \frac{\alpha(1 + d(\mathtt{nx}))^{-\beta} \left[1 - \alpha(2 + d(\mathtt{nx}))^{-\beta}\right]^2}{1 - \alpha(1 + d(\mathtt{nx}))^{-\beta}} \times \frac{\mathtt{nleaf}(\mathcal{T})}{\mathtt{nnog}(\mathcal{T}^\star)}
$$
$$
\times \tau^{-1} \times \left(\frac{P_{\mathtt{nxl}} P_{\mathtt{nxr}}}{P_{\mathtt{nx}}}\right)^{-\frac{1}{2}} \times \exp\left\{\frac{\Theta_{\mathtt{nxl}}^2}{2P_{\mathtt{nxl}}} + \frac{\Theta_{\mathtt{nxr}}^2}{2P_{\mathtt{nxr}}} - \frac{\Theta_{\mathtt{nx}}}{2P_{\mathtt{nx}}}\right\} \tag{B1}
$$
$$
\times \frac{p(\mathtt{rule}|\mathcal{T}^\star, \mathcal{D}^\star)}{q(\mathtt{rule}|\mathcal{T}, \mathcal{D})}.
$$

The terms in the first two lines of Equation (B1) respectively compare the complexity (i.e., depth) and overall fit to the partial residual $\boldsymbol{R}$ of $(\mathcal{T}^\star, \mathcal{D}^\star)$ and $(\mathcal{T}, \mathcal{D})$. More specifically, the term in second line tends to be larger than one whenever splitting the tree at $\mathtt{nx}$ along $\mathtt{rule}$ yields a better fit to the current partial residual than not splitting the tree at $\mathtt{nx}$. The term in the final line is the ratio between the prior and proposal probability of drawing the rule $\mathtt{rule}$. When we draw $\mathtt{rule}$ from the prior, the MH acceptance probability depends only on terms that compare the fit and complexity of the two trees. However, if the proposal distribution is much more sharply concentrated around $\mathtt{rule}$ than the prior, this term will artificially deflate the acceptance probability.

**Prune move**. Suppose instead that we form $(\mathcal{T}^\star, \mathcal{D}^\star)$ by removing leafs `nxl` and `nxr` and turning their common parent `nx` into a leaf. Let `rule` denote the rule associated with `nx` in $(\mathcal{T}, \mathcal{D})$. The acceptance probability of a prune move is

$$\alpha(\mathcal{T}, \mathcal{D} \to \mathcal{T}^\star, \mathcal{D}^\star) = \frac{1 - \alpha(1 + d(\texttt{nx}))^{-\beta}}{\alpha(1 + d(\texttt{nx}))^{-\beta} \left[1 - \alpha(2 + d(\texttt{nx}))^{-\beta}\right]^2} \times \frac{\texttt{nnog}(\mathcal{T})}{\texttt{nleaf}(\mathcal{T}^\star)}$$
$$\times \tau \times \left(\frac{P_{\texttt{nx}}}{P_{\texttt{nxl}} P_{\texttt{nxr}}}\right)^{-\frac{1}{2}} \times \exp\left\{\frac{\Theta^2_{\texttt{nx}}}{2P_{\texttt{nx}}} - \frac{\Theta^2_{\texttt{nxl}}}{2P_{\texttt{nxl}}} - \frac{\Theta^2_{\texttt{nxr}}}{2P_{\texttt{nxr}}}\right\} \qquad \text{(B2)}$$
$$\times \frac{q(\texttt{rule}|\mathcal{T}^\star, \mathcal{D}^\star)}{p(\texttt{rule}|\mathcal{T}, \mathcal{D})}.$$

## C Hyperparameter Sensitivity Analysis

As noted in Section 3, obliqueBART depends on several hyperparameters. In this section, we perform a sensitivity analysis using the benchmark datasets from the experiments in Section 4. Specifically, we investigate the sensitivity to the hyperparameters for the sparcity parameter $\theta$, which regulates the number of non-zero entries in $\phi$. We also assess the sensitivity to the number of trees, $M$, on the performance of obliqueBART.

**Sensitivity to $\theta$**

We ran obliqueBART with multiple combinations for the hyperparameters on the Beta $(a_\theta, b_\theta)$ prior on $\theta$. Specifically, we ran obliqueBART with $a_\theta \in \{1, p, M\}$, with $M = 200$, and $b_\theta$ such that $\mathbb{E}[\theta] \in \{\frac{1}{p}, \frac{2}{p}, \frac{\sqrt{p}}{p}, \frac{1}{2}\}$. Holding $b_\theta$ constant, increasing $a_\theta$ allows obliqueBART to be less sensitive to the data. Recall that the conjugate update for $\theta$ is

$$\theta | \boldsymbol{y}, \mathcal{E}, \sigma^2 \sim \text{Beta} \left(a_\theta + n_\phi, b_\theta + z_\phi\right)$$

So, when $a_\theta$ is small, the effect of $n_\phi$ (resp. $z_\phi$), the number of non-zero (resp. zero) entries, is more substantial, compared to when $a_\theta$ is large. The other hyperparameter, $b_\theta$ is chosen to control the sparsity of $\phi$, and we test the performance of obliqueBART when the prior mean for $\theta$ is dependent and independent of $p$. Figure C8 depicts boxplots for out of sample SMSE across all splits and datasets for the difference combinations of $(a_\theta, b_\theta)$.

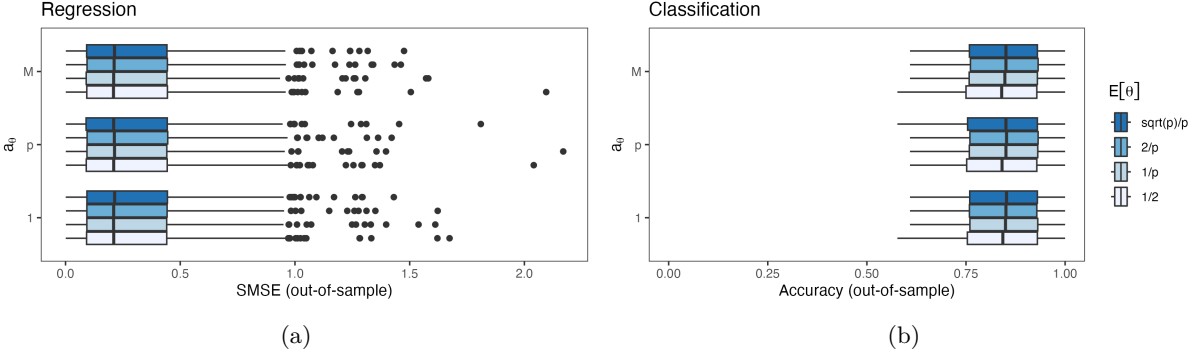

Figure C8: ObliqueBART's SMSE (a) and accuracy (b) across all splits and datasets, compared at different combinations of $(a_\theta, b_\theta)$. Models with lower SMSE's and higher accuracies are preferred.

ObliqueBART seems insensitive to the choices of $a_\theta$ and $b_\theta$. For the vast majority of datasets, predictive accuracy is very similar. We note that, generally, setting $b_\theta$ so that $\mathbb{E}[\theta]$ is dependent on $p$ tended to produce better results than $\mathbb{E}[\theta] = 1/2$. We move forward with $a_\theta = M$ and $b_\theta$ set so that $\mathbb{E}[\theta] = 2/p$ for the rest of the experiments and recommend this as the default setting for obliqueBART.

**Sensitivity to $M$**

Fixing $a_\theta$ to $M$ and $b_\theta$ so that $\mathbb{E}[\theta] = 2/p$, we ran obliqueBART with $M \in \{10, 25, 100, 200\}$. Figure C9 shows the resulting boxplots for out of sample SMSE across all splits and benchmark datasets. We find that $M = 200$ generally produced the best predictive accuracy, although the accuracies were quite similar, but slightly worse, for $M = 100$.

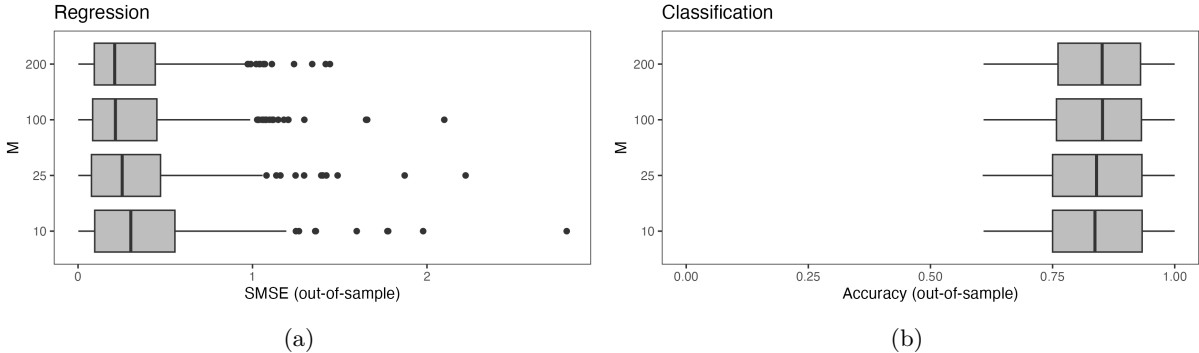

(a)                      (b)

Figure C9: ObliqueBART's SMSE (a) and accuracy (b) across all splits and datasets, compared at different levels of $M$. Models with lower SMSE's and higher accuracies are preferred.

