# OpenReview forum: "Oblique Bayesian Additive Regression Trees"
_TMLR — Accepted by TMLR_

### Review · Reviewer_47xj · 2025-02-08

**Summary Of Contributions:**

This paper proposes a method for constructing a BART model that performs oblique splitting and evaluates its effectiveness through several experiments.

**Audience:**

No

**Claims And Evidence:**

No

**Requested Changes:**

- The experiments primarily compare the proposed method with axis-aligned models, but this does not seem to differ significantly from the claims made in prior work on oblique trees. For example, nowhere in the paper is a direct comparison made between typical oblique trees and obliqueBART. Furthermore, the discussion on BART-specific characteristics, such as uncertainty estimation, is lacking. As a result, the current presentation does not provide a strong motivation for using oblique BART over a standard oblique tree.

- Due to sampling, proposed obliqueBART is likely to have a higher computational cost compared to simple oblique trees. This aspect should be evaluated. Including such a comparison is essential to justify why obliqueBART is preferable to standard oblique trees.

- Compared to simple axis-aligned decision trees, oblique splitting is more sensitive to feature scaling and handling of missing values, as these factors directly impact performance. However, the paper does not mention how these preprocessing steps were handled.

- Although the Appendix discusses the hyperparameters of oblique BART, the benchmark methods do not tune hyperparameters. This makes the performance comparison unfair.

- Minor points:

  - The introduction and other sections repeatedly emphasize performance improvements using numerical percentages, but this is not a common practice and should be avoided.

  - The quality of figures presenting experimental results is insufficient. For example, Figure 4 lacks error evaluation, and the x-axis of the left-side plot in Figure 4 appears to use decimal notation with an upper limit of $\pi/4$, which is unintuitive and difficult to interpret.

  - In Section 2, there is a statement suggesting that GBT requires trees to have a fixed shallow depth. However, this is not a general constraint. There are several instances where conventions and specifications seem to be conflated.

**Strengths And Weaknesses:**

Generalizing BART to use oblique splitting may seem like a straightforward extension, yet it has not been explored before. This paper may have the potential to contribute an important piece to the machine learning community. Additionally, the paper provides intuitive visualizations using figures and a thorough introduction to both BART and oblique splitting, making it accessible even to readers unfamiliar with these concepts. The approach described in the paper for extending BART with oblique splitting is straightforward and does not introduce significant technical novelty. However, since TMLR’s evaluation criteria do not primarily emphasize technical contribution and significance, this is not a major concern.

On the other hand, I see several issues with the presentation and experimental demonstration. In particular, the discussion on aspects unique to BART is rather limited. The experiments compare a model using oblique splitting with one using axis-aligned splitting, which is a major limitation. If this were the main focus, traditional oblique decision trees could be used instead. Therefore, it is crucial to clearly present the advantages of using oblique trees specifically within BART. I will elaborate on this point in detail in the Requested Changes section.

---

> ### Author Response · Authors · 2025-03-04
> **Thanks!**
>
> **Nowhere in the paper is a direct comparison made between typical oblique trees and obliqueBART.**
>
> In our original experiments, we focused on competitors that were (i) easily implemented in R and (ii) capable of performing both regression and classification. In our initial review of the literature, we found that several oblique tree implementations were designed only for classifications and many were not compatible with updated versions of R (and some had been removed from CRAN). We have gone back and conducted a wider search for oblique tree methods. We have included a new section in the manuscript (Section 4.2.2) that compares  obliqueBART to the following implementations of oblique trees and forests:
> [RPEnsemble](https://cran.r-project.org/web/packages/RPEnsemble/index.html); [rotationForest](https://cran.r-project.org/web/packages/rotationForest/rotationForest.pdf); [ODRF](https://cran.r-project.org/web/packages/ODRF/index.html); [aorsf](https://cran.r-project.org/web/packages/aorsf/index.html).
>
> **Due to sampling, the proposed obliqueBART is likely to have a higher computational cost compared to simple oblique trees. This aspect should be evaluated.**
>
> While BART and obliqueBART typically take longer to compute compared to non-Bayesian tree ensembles, their ability to quantify uncertainty often make the increased computation time worthwhile. We have added this note in the paper, and compared the computational cost of obliqueBART to standard axis-aligned BART in Section 4.2.1.
>
>
> **Compared to simple axis-aligned decision trees, oblique splitting is more sensitive to feature scaling and handling of missing values, as these factors directly impact performance. However, the paper does not mention how these preprocessing steps were handled**
>
> In our experiments, continuous variables were scaled to -1 to 1, and observations with missing values were dropped before we fit any models to the data. We have added a sentence mentioning the preprocessing steps to Section 4.2 of the manuscript.
>
> **Although the Appendix discusses the hyperparameters of oblique BART, the benchmark methods do not tune hyperparameters. This makes the performance comparison unfair.**
>
> In our experiments, *all methods*, including obliqueBART, were run with the package defaults. Importantly, we did not tune the hyperparameters of obliqueBART in a dataset-specific fashion. Instead, we ran it with the default recommendations from the paper. So, in that sense, all methods were on a level playing field in these experiments. The purpose of the sensitivity analysis in Appendix was to show that obliqueBART is not particularly sensitive to the choice of hyperparameters.
>
> **The introduction and other sections repeatedly emphasize performance improvements using numerical percentages, but this is not a common practice and should be avoided**
>
> Many papers in the literature use different evaluation metrics, and do not report improvements in a consistent fashion. For example, Breiman (2001) compares error rates on an individual dataset to dataset level. Bertsimas (2017) reports percentage gains in terms of out-of-sample accuracy, while Rainforth and Wood (2017) report that their method reduces the number of misclassifications by a factor of 28.7%. Blaser and Fryzlewicz (2016) compared random rotation ensembles to other methods by their average rank.  Tomita et. al (2019) and  Li et. al (2022) report Cohen’s kappa, and Li et. al (2022) also reports AUC. We tried to report these gains in a consistent manner using numerical percentages when the original papers provided enough information to do so.
>
> **The quality of figures presenting experimental results is insufficient. For example, Figure 4 lacks error evaluation, and the x-axis of the left-side plot in Figure 4 appears to use decimal notation with an upper limit of , which is unintuitive and difficult to interpret.**
>
> We have added error bars in Figure 4. We have also changed the decimal notation in the left side of the figure to units of $\pi$, which may help interpretation of the angle of rotation.
>
> **In Section 2, there is a statement suggesting that GBT requires trees to have a fixed shallow depth. However, this is not a general constraint. There are several instances where conventions and specifications seem to be conflated**
>
> Thank you for pointing this out. We have revised our manuscript to say that "when running GBT, practitioners conventionally fix the tree to have shallow depth"

---

> > ### Comment · Reviewer_47xj · 2025-03-12
> >
> > Thank you for revising your paper. In particular, I appreciate the comparison with the oblique tree ensemble.
> >
> > Personally, I feel that comparing results with the default hyperparameters of the libraries used as benchmarks may not carry much significance in terms of performance comparison, but aside from that, I was able to understand your responses.

---

> ### Comment · Action_Editor_RARC · 2025-03-12
> **Official Comment by AE**
>
> The use of default hyperparameters sounds problematic to me as well.
> TMLR requires the submission needs to be supported by accurate, convincing and clear evidence.
>
> https://jmlr.org/tmlr/acceptance-criteria.html
>
> In the current state, it would not be rigid enough to conclude that "oblique BART was competitive with — and sometimes much better than — those methods".
> The hyperparameters need to be set by some appropriate ways for fair comparisons.

---

> > ### Author Response · Authors · 2025-03-14
> > **Hyperparameter tuning**
> >
> > Thank you for your comments and suggestions regarding model comparisons.
> >
> > In the interest of conducting a fair and robust study to better support our claims, we would like to request more time to re-run our experiments with tuned competitors. We have already updated our code to tune hyperparameters using the same settings and process as Chipman et al. (2010) in the original BART paper. But it will take us some more time to finish running on all training/testing splits on all benchmark datasets.
> >
> > Given the high degree of overlap in benchmark datasets between our manuscript and Chipman et al. (2010), we anticipate obtaining results similar to those shown in Figure 2 of that paper: tuned versions of RF and GBT perform similarly to axis-aligned BART run with default hyperparameters.
> >
> > We anticipate that we will finish running the experiments, updating the figures & tables in the manuscript, and making any necessary revisions to the text by March 21.

---

> > > ### Comment · Action_Editor_RARC · 2025-03-14
> > > **Re: Hyperparameter tuning**
> > >
> > > Extra one week for the additional experiments will not be a problem.
> > > I believe this is an essential step needed for the paper decision.
> > > I will talk to the TMLR editors and will let you know if there is any problem.

---

> > > > ### Comment · Reviewer_47xj · 2025-03-21
> > > >
> > > > Thank you for conducting the experiment with cross-validation.
> > > >
> > > > Looking at the updated Figure 6 and Table A5, I find the results somewhat puzzling. For example, in Table A5, for the edu dataset, the SMSE values for the benchmark methods are 0.795 and 0.806, whereas the SMSE for the proposed method is 0.02.
> > > > Similarly, for the labor dataset, the SMSE of the proposed method is about one-fourth that of the benchmark methods—again, a very large difference. These differences appear to have a significant impact on the results reported in Figure 6 and elsewhere.
> > > >
> > > > Before the revision, I thought that the parameter settings might be the main reason for these differences. However, do the authors have any thoughts on other possible explanations?
> > > >
> > > > If the performance gap between methods were around 10 percent, I would find it reasonable. However, such a dramatic difference feels unintuitive to me, and I would appreciate it if the authors could help clarify the reasons behind it.

---

> > > > > ### Author Response · Authors · 2025-03-21
> > > > > **Performance gaps**
> > > > >
> > > > > Thank you for your careful review of our empirical results and for raising this issue. After reviewing our experiment code, we identified a small bug in how we preprocessed categorical predictors before fitting the competing oblique methods. We corrected this issue, re-ran the experiments, and have updated our figures and commentary in Section 4.2.2. Though the figures and the values in the table have changed, our qualitative conclusions did not: obliqueBART is still slightly better at regression and is competitive for classification than its oblique competitors.
> > > > >
> > > > > We have updated the dataset-by-dataset performances in Tables A5 and A6. For the *edu* dataset, there is no longer such a stark difference in performance between obliqueBART and ODRF and aorsf. However, even after correcting the pre-processing of categorical predictors, obliqueBART appears much better than ODRF and aorsf on the *labor* dataset. On further inspection, we found that this dataset contains a categorical predictor with a huge number of levels (the dataset contains repeated observations from several subjects; the variable in question identifies the subject). We believe the gap in performance on this dataset stems from how the different methods handle categorical features.
> > > > >
> > > > > Briefly, both ODRF and aorsf one-hot encoded this variable, creating many additional binary features, one for each level of the categorical variable. Oblique decision rules built with these features generally assign one or two levels to the left branch of the tree and everything else to the right (or vice versa). In other words, the trees in the ODR and aorsf ensembles partition the categorical levels by recursively ``removing a few at a time''. ObliqueBART, in contrast, considers a wider range of partitions of categorical levels. Generally speaking, when a tree in the obliqueBART ensemble splits on a categorical feature, it assigns about half the available levels to the left branch and the remaining half to the right branch. In this way, obliqueBART is able to "borrow strength" across categories more flexibly (and usually more effectively) than the other methods. The results on this dataset are consistent with those of [Deshpande (2024)](https://www.tandfonline.com/doi/full/10.1080/10618600.2024.2431072), who demonstrated the superiority of the categorical decision rule prior we employ here over one-hot encoding.

---

> > > > > > ### Comment · Reviewer_47xj · 2025-03-22
> > > > > >
> > > > > > Thank you for your answer. I have read your response.

---

### Review · Reviewer_oFXc · 2025-02-17

**Summary Of Contributions:**

This research proposes a variant of Bayesian Additive Regression Trees (BART) that partitions the feature space using non-axis-aligned hyperplanes, termed as oblique BART. The authors also propose a sampling method from the posterior distribution of oblique BART. The model can handle both continuous and categorical predictor variables and is applicable to both regression and classification problems. On synthetic datasets where oblique BART is intuitively expected to perform well, it demonstrates superior performance compared to conventional methods as expected. In terms of predictive performance on benchmark datasets, oblique BART shows slightly better results than existing methods.

**Audience:**

Yes

**Broader Impact Concerns:**

Not applicable.

**Claims And Evidence:**

Yes

**Requested Changes:**

## Critical

* Add more detailed explanations for the derivation of equations (1), (2), and (3)
  * The derivation of equations (1), (2), and (3) appears to have some logical gaps. Please include the complete transformations based on conditional independence derived from model assumptions without omissions. For instance, I believe equation (1) uses conditional independence of $(\mathcal{T}, \mathcal{D})$ and $(\mathcal{E}^-, \sigma^2)$ given $\theta$ after transformations such as:
$$p(\mathcal{T}, \mathcal{D}, \mathcal{M} | \boldsymbol{y}, \mathcal{E}^-, \sigma^2, \theta) \propto p( \boldsymbol{y} | \mathcal{T}, \mathcal{D}, \mathcal{M}, \mathcal{E}^-, \sigma^2, \theta) p(\mathcal{M} | \mathcal{T}, \mathcal{D}, \mathcal{E}^-, \sigma^2, \theta) p(\mathcal{T}, \mathcal{D} | \mathcal{E}^-, \sigma^2, \theta).$$
Please include such complete transformations for equations (2) and (3) as well.
* Address the following descriptive deficiencies:
  * Title: Capitalize the first letter of each word appropriately.
  * Page 2, 3rd line from bottom: $\mathcal{M}$ is not explicitly defined. Is it a set of $\mu_l$?
  * Page 5, line 7: $\mathrm{Inv. Gamma}(3/2, 3\lambda/2)$ should include $\nu$.
  * Sometimes, the word "obliqueBART" is not properly capitalized:
    * Page 5, 2nd line from bottom
    * Page 6, 10th line from bottom
    * Page 11, line 12
  * Page 6, 17th line from bottom: $\gamma_p$ should be $\gamma_{ p_\mathrm{cont} }$
  * Before equation (1), either explicitly state that you are considering the case where $(\mathcal{T}_m, \mathcal{D}_m, \mathcal{M}_m) = (\mathcal{T}, \mathcal{D}, \mathcal{M})$, or add subscript $m$ to $\mathcal{T}$, $\mathcal{D}$, $\mathcal{M}$ in equations (1), (2), and (3).
  * Page 9, 10th line from bottom: Figure 2a should be Figure 2d.
  * Equation (B1): $\mathcal{D}^\star$ is missing in the numerator and $\mathcal{D}$ is missing in the denominator of the last term.
  * Equation (B2): There is an unnecessary bracket $]$ at the end of the numerator in the first term.
  * Equation (B2): The superscripts $\star$ on $\mathcal{T}^\star$ and $\mathcal{D}^\star$ in the denominator of the last term are unnecessary.

## Personal Suggestion

The following modifications are not mandatory for acceptance. Please consider them if you find them agreeable.

### **Regarding the derivation order of equations (1), (2), and (3)**

In my understanding, equations (2) and (3) are derived by comparing the following two equations, where the first equation should correspond to equation (1) in the manuscript:
$$\begin{align*}
  p(\mathcal{T}, \mathcal{D}, \mathcal{M} | \boldsymbol{y}, \mathcal{E}^-, \sigma^2, \theta) &\propto p( \boldsymbol{y} | \mathcal{T}, \mathcal{D}, \mathcal{M}, \mathcal{E}^-, \sigma^2, \theta) p(\mathcal{M} | \mathcal{T}, \mathcal{D}, \mathcal{E}^-, \sigma^2, \theta) p(\mathcal{T}, \mathcal{D} | \mathcal{E}^-, \sigma^2, \theta) \\\\
  p(\mathcal{T}, \mathcal{D}, \mathcal{M} | \boldsymbol{y}, \mathcal{E}^-, \sigma^2, \theta) &= p( \mathcal{T}, \mathcal{D} | \boldsymbol{y}, \mathcal{E}^-, \sigma^2, \theta) p(\mathcal{M} | \mathcal{T}, \mathcal{D}, \boldsymbol{y}, \mathcal{E}^-, \sigma^2, \theta)
\end{align*}$$
While this approach is valid, it might be more straightforward to first derive equation (3) directly using basic properties of probability and the definition of the model:
$$\begin{align*}
&p(\mathcal{M} | \mathcal{T}, \mathcal{D}, \boldsymbol{y}, \mathcal{E}^-, \sigma^2, \theta) \\\\
&\propto p(\boldsymbol{y} | \mathcal{T}, \mathcal{D}, \mathcal{M}, \mathcal{E}^-, \sigma^2, \theta)p(\mathcal{M} | \mathcal{T}, \mathcal{D}, \mathcal{E}^-, \sigma^2, \theta) \\\\
&= (2\pi)^{-\frac{n}{2}} \sigma^{-n} \exp \left( -\frac{1}{2 \sigma^2} \sum_{i=1}^n r_i^2 \right) \prod_{l: \mathrm{leaf}} \frac{1}{\sqrt{2\pi \tau^2}} \exp \left( -\frac{1}{2} P_l (\mu_l - P_l^{-1} \Theta_l )^2 + \frac{ \Theta_l^2 }{ 2P_l } \right)  \qquad ( * ) \\\\
&\propto \prod_{l: \mathrm{leaf}} \exp \left( -\frac{1}{2} P_l (\mu_l - P_l^{-1} \Theta_l )^2 \right)
\end{align*}$$
Then, by substituting $( * )$ and using the fact that $\int \exp \left( -\frac{1}{2} P_l (\mu_l - P_l^{-1} \Theta_l )^2 \right) \mathrm{d} \mu_l = \sqrt{\frac{2 \pi}{P_l}}$, equation (2) can be derived:
$$\begin{align*}
&p( \mathcal{T}, \mathcal{D} | \boldsymbol{y}, \mathcal{E}^-, \sigma^2, \theta) \\\\
&\propto p(\mathcal{T}, \mathcal{D} | \mathcal{E}^-, \sigma^2, \theta) \int p(\boldsymbol{y} | \mathcal{T}, \mathcal{D}, \mathcal{M}, \mathcal{E}^-, \sigma^2, \theta)p(\mathcal{M} | \mathcal{T}, \mathcal{D}, \mathcal{E}^-, \sigma^2, \theta) \mathrm{d} \mathcal{M} \\\\
&= p(\mathcal{T}, \mathcal{D} | \theta) \int (2\pi)^{-\frac{n}{2}} \sigma^{-n} \exp \left( -\frac{1}{2 \sigma^2} \sum_{i=1}^n r_i^2 \right) \prod_{l: \mathrm{leaf}} \frac{1}{\sqrt{2\pi \tau^2}} \exp \left( -\frac{1}{2} P_l (\mu_l - P_l^{-1} \Theta_l )^2 + \frac{ \Theta_l^2 }{ 2P_l } \right) \mathrm{d}\mathcal{M} \\\\
&\propto p(\mathcal{T}, \mathcal{D} | \theta) \prod_{l: \mathrm{leaf}} \tau^{-1} P_l^{-\frac{1}{2}} \exp \left( \frac{ \Theta_l^2 }{ 2P_l } \right)
\end{align*}$$
Note that the second equality uses the conditional independence of $(\mathcal{T}, \mathcal{D})$ and $(\mathcal{E}^-, \sigma^2)$ given $\theta$.

### **Regarding the distinction between conventional and proposed methods**

While the current manuscript clearly states that the originality of the proposed method lies solely in the generalization of grow proposals, to make the distinction between conventional and proposed methods even clearer, you might consider moving the following sections into Section 2.2.2:

* Section 3.2.1 (excluding the Efficient decision tree update part)
* Section 3.2.2

**Strengths And Weaknesses:**

## Strengths

* The proposed model and algorithm demonstrate novelty, as there has not been a BART variant with oblique splits before.
* The method of oblique partitioning and prior distributions for the parameters used in partitioning are clearly defined and explained for both continuous and discrete predictor variables.
* The sampling method from the posterior distribution is technically correct.
* The model shows expected superior performance on synthetic datasets where oblique BART is intuitively well-suited.
* Guidelines for hyperparameter tuning are provided.

## Weaknesses

* There are numerous deficiencies in the description.
* There is room for improvement in the clarity of the derivation of equations (1), (2), and (3).
* The performance improvement on benchmark datasets is not substantial. However, since significance is not heavily emphasized in TMLR, this is not a major concern.

---

> ### Author Response · Authors · 2025-03-04
> **Thanks!**
>
> **Before equation (1), either explicitly state that you are considering the case where $(\mathcal{T}, \mathcal{D}, \mathcal{M}) = (\mathcal{T}_m, \mathcal{D}_m, \mathcal{M}_m)$, or add subscript $m$ to $\mathcal{T}, \mathcal{D}, \mathcal{M}$. Add more detailed explanations for the derivation of equations (1)--(3)**
>
> We have revised the manuscript to clarify we are considering the case where $(\mathcal{T}, \mathcal{D}, \mathcal{M}) = (\mathcal{T}_m, \mathcal{D}_m, \mathcal{M}_m)$.
>
> We have updated Equation 1 in the revised manuscript to make the conditional independence assumptions explicit. We have also added a line stating that the Equation 2 is obtained by integrating out the $\mu_{\ell}$'s from Equation 1. While we appreciate the reviewer's detailed derivation, we note that these calculations are relatively standard normal-normal conjugate updates (see, e.g., Equations 2.11 & 2.12, and Section 2.5 of the 3rd edition of *Bayesian Data Analysis*).
>
> **$\mathcal{M}$ is not explicitly defined. Is it a set of $\mu_{\ell}$'s?**
>
> Yes. We apologize for this oversight and have explicitly defined $\mathcal{M}$ in our revision.
>
> **Reorganizing the presentation**
>
> We appreciate the reviewer's suggestion to move parts of Section 3.2 into Section 2.2. While we see the logic behind fully presenting the original BART model in Section 2 and only focusing on our new additions in Section 3, we prefer to present obliqueBART fully.
>
> **Typographical comments and issue with Appendix equations**
> We appreciate the referee for pointing out several grammatical, typographical errors, and issues with the equations in the Appendix. We have corrected them in our revised manuscript

---

> > ### Comment · Reviewer_oFXc · 2025-03-12
> >
> > Thank you for revising your paper.
> > All my concerns except the title capitalization are addressed; "Oblique Bayesian additive regression trees" should be "Oblique Bayesian Additive Regression Trees."
> > Please fix it in the camera-ready version.
> >
> > P.S. In my first look, I misunderstood that Eqs. (2) and (3) are derived from Eq. (1) at the same time by comparing the right-hand side of Eq. (1) and the product of Eqs. (2) and (3), and it looked a little tricky to me. Since the revised description explicitly stated only Eq. (2) is  derived by marginalizing Eq. (1) first, it became easy to understand, at least for me. Thank you.

---

### Review · Reviewer_wbnm · 2025-02-21

**Summary Of Contributions:**

The authors develop an oblique-split version of the BART
algorithm. Since this is a Bayesian approach, this requires choosing a
prior on the oblique decision rules. The method is presented (after
suitable background is given) and empirical results presented.

**Audience:**

Yes

**Claims And Evidence:**

Yes

**Requested Changes:**

What I find missing here is an empirical comparison where there is
*no* random rotation of the data. My assumption is that oblique-BART
might do worse than regular BART (and perhaps also other comparators),
but I don't know for sure. Please include that.

**Strengths And Weaknesses:**

This is a conservative extension of BART. For example, the priors on T
and M as Chipman et al. This seems a sensible approach: BART is a
successful approach so it makes sense to 'just' add in the capability
to have oblique decision rules and see what happens.

The paper is clearly written and not a hard read. There are no
theoretical results, just a description and justification for the
design of oblique BART and empirical results. A useful finding from
the empirical work is an indication that the hierarchical
spike-and-slab prior allows oblique BART to adjust the ratio between
axis-aligned and oblique rules.

There are no big surprises in the empirical results. Naturally when
true decision boundaries are oblique a model which allows oblique
decision boundaries has an advantage. It was somewhat interesting that
randomly rotated versions of BART, RF, ERT, and XGB did fairly
badly. On the benchmark datasets oblique-BART's performance "is generally not
significantly worse than BART's and can sometimes be substantially
better." It's good to see the authors not overselling their algorithm.

I think a future version of this paper might well be suitable for
publication in TMLR: I do not have a problem with the "incrementality"
of the contribution. Most papers make such a contribution (or none at
all). Some might view the empirical results as mainly negative (for
example oblique BART had the worst average classification accuracy of
all methods), but the point is that we have useful results, whether we
choose to interpret them as negative or otherwise.

SMALL POINTS

in (1) etc "leafs" -> "leaves"

"Such proposals ... receives" ->
"Such proposals ... receive"

Figure 4a shows how axis-aligned BART and random rotation.. ->
Figure 4 shows how axis-aligned BART and random rotation..

p11
it's average accuracy ->
its average accuracy

My guess (I should not have to guess) is that the "average" SMSE of
0.296 mentioned on p11 is mean SMSE and the vertical line on the plots
in Fig 5 indicates the median (which would be a standard for a
boxplot). The vertical line for XGB is to the left of that for ERT
even though its "average" SMSE (0.342) is higher than that of ERT
(0.332). It would be good to report median values in the text.

---

> ### Author Response · Authors · 2025-03-04
> **Thanks!**
>
> **My guess (I should not have to guess) is that the "average" SMSE ... is mean SMSE and the vertical line on the plots in Fig 5 indicates the median. It would be good to report median values (of SMSE) in the text**
>
> You are correct that "average" here refers to the mean SMSE and the vertical lines in the figure indicate the median SMSE. We have added a sentence reporting the median values to revised manuscript.
>
> **Empirical comparison where there is *no* random rotation of the data**
>
> In Section 4.1, we compared obliqueBART with axis-aligned BART with varying numbers of random rotations, including 0 random rotations (gray line). Section 4.2.1 and Figure 5 detail the comparisons with other axis-aligned tree ensembles on the benchmarks, with no random rotations of the data.
>
> **Grammar and typographical issues**
>
> Thank you for pointing out the grammatical and typographical errors. We have fixed these in our revised manuscript.

---

### Author Response · Authors · 2025-03-19
**Updated experimental results**

Thank you for the extension. We have re-run our experiments with all of the competing methods (axis-aligned and oblique) using tuned hyperparameters. For each non-BART-based method, we tuned hyperparameters using 5-fold cross-validation on each training dataset. Though the performance of the competing methods generally improved, the qualitative results are very similar to our initialy experiments. We find that obliqueBART with our recommended default hyperparameter settings remains competitive in the classification setting and performs slightly better in the regression setting compared to the tuned methods. We have updated the relevant figures, tables, and exposition in Section 4.2 and Appendix A to reflect the results of our updated experiments.

---

### Decision · Action_Editor_RARC · 2025-04-02

**Recommendation:** Accept with minor revision

**Comment:**

In this paper, the authors proposed a Bayesian Additive Regression Trees (BART) method that incorporates oblique splitting. While the reviewers acknowledged that the techniques used are a combination of existing approaches, they agreed that this work presents the first instance of BART considering oblique splitting, thus considered novel.
However, the reviewers also noted that the significance of this research is somewhat limited. Through discussion, the authors addressed concerns by improving the paper's readability and notation, and by updating the experiments to provide further evidence of the proposed method's effectiveness. Based on these revisions, the paper is now considered suitable for acceptance by TMLR.

The following are the final revisions requested by the reviewers for the paper's acceptance:

* Section 4: "Generally speaking, Generally speaking"
* "smallest classification accuracy" -> "lowest classification accuracy"
* Figure 4 should be more self-contained. This means a longer caption which repeats the information about what information Fig 4 is showing. This information is currently in various places in the main text. Also is the information in Fig 4 for some particular choice of n, or are the RMSEs for different values of n combined?
* In Table A.2, please ensure consistency in the capitalization of terms.
* Reproducible code has not yet been released.

**Audience:**

Oblique splitting is one of the basic techniques in decision trees, and Bayesian Additive Regression Trees is one type of the boosting methods. This paper considers the combination of them, and thus it would be of interest of a part of the machine learning community.

**Claims And Evidence:**

In this paper, the authors proposed a Bayesian Additive Regression Trees (BART) method that incorporates oblique splitting. The paper provides a detailed derivation of the specific techniques used, and the experiments demonstrate the effectiveness of the proposed method. Although the initial submission had some issues with the experimental setups, through discussion, the authors updated the experiments, which has made the reported results more convincing.